

# FESOM-C: coastal dynamics on hybrid unstructured meshes

Alexey Androsov[AWI, IO], Vera Fofonova[AWI], Ivan Kuznetsov[HZG], Sergey Danilov[AWI, IAP, J],
Natalja Rakowsky[AWI], Sven Harig[AWI], and Karen Helen Wiltshire[AWI]

[AWI]Alfred Wegener Institute for Polar and Marine Research, Bremerhaven, Germany
[HZG]Institute of Coastal Research, Helmholtz-Zentrum Geesthacht, Geesthacht, Germany
[IO]Shirshov Institute of Oceanology, Moscow, Russia
[IAP]A. M. Obukhov Institute of Atmospheric Physics, Moscow, Russia
[J]Jacobs University, Bremen, Germany

*Correspondence to:* Alexey Androsov (alexey.androsov@awi.de)

**Abstract.** We describe FESOM-C, the coastal branch of the Finite-volumE Sea ice – Ocean Model (FESOM2), which shares with FESOM2 many numerical aspects, in particular, its finite-volume cell-vertex discretization. Its dynamical core differs by the implementation of time stepping, the use of terrain-following vertical coordinate and formulation for hybrid meshes composed of triangles and quads. The first two distinctions were critical for coding FESOM-C as an independent branch. The
5 hybrid mesh capability improves numerical efficiency, since quadrilateral cells have fewer edges than triangular cells. They do not suffer from spurious inertial modes of the triangular cell-vertex discretization and need less dissipation. The hybrid mesh capability allows one to use quasi-quadrilateral unstructured meshes, with triangular cells included only to join quadrilateral patches of differt resolution or instead of strongly deformed quadrilateral cells. The description of the model numerical part is complemented by test cases illustrating the model performance.

## 10 1 Introduction

Many practical problems in oceanography require regional focus on coastal dynamics. Although global ocean circulation models formulated on unstructured meshes may in principle provide local refinement, such models are as a rule based on assumptions that are not necessarily valid in coastal areas. The limitations on dynamics coming from the need to resolve thin layers, maintain stability for sea surface elevations comparable to water layer thickness or simulate the processes of wetting and
15 drying make numerical approaches traditionally used in coastal models different from those used in large-scale models. For this reason, combining a coastal and large-scale functionality in a single unstructured-mesh model, although possible, would still imply a combination of different algorithms and physical parameterizations. Furthermore, on unstructured meshes, numerical stability of open boundaries, needed in regional configurations, sometimes requires to mask certain terms in motion equations close to open boundaries. This would be an unnecessary complication for a large-scale unstructured-mesh model which is as a
20 rule global.

The main goal of the development described in this paper was to design a tool, dubbed FESOM-C that is close to FESOM2 (Danilov et al., 2017) in its basic principles, but can be used as a coastal model. Its routines handling the mesh infrastructure are derived from FESOM2. However, the time stepping, vertical discretization, and particular algorithms, detailed below,





are different. FESOM-C relies on terrain-following vertical coordinate (vs. the Arbitrary Lagrangian Eulerian (ALE) vertical coordinate of FESOM2), but does a step further with respect to the mesh structure. It is designed to work on hybrid meshes composed of triangles and quads. Some decisions, such as for example, the lack of the ALE at the present stage, are only motivated by the desire to keep the code as simple as possible through the initial phase of its development and maintenance.

The code is based on the cell-vertex finite-volume discretization, same as FESOM2 (Danilov et al., 2017) and FVCOM (Chen et al., 2003). It places scalar quantities at mesh vertices and the horizontal velocities at cell centroids.

Our special focus is on using hybrid meshes. In essence, the capability of hybrid meshes is build in finite-volume method. Indeed, computations of fluxes are commonly implemented as cycles over edges, and the edge-based infrastructure is immune to the polygonal type of mesh cells. However, because of staggering, it is still convenient to keep some computations on cells,

which then depend on the cell type. Furthermore, high-order transport algorithms might be sensitive to the cell geometry too. We limit the allowed polygons to triangles and quads. Although there is no principal limitation on the polygon type, triangles and quads are versatile enough in practice for the cell-vertex discretization. Our motivation of using quads is two-fold (Danilov and Androsov, 2015). First, quadrilateral meshes have 1.5 times fewer edges than triangular meshes, which speeds up computations because cycles over edges become shorter. The second reason is the intrinsic problem of the triangular cell-

vertex discretization — the presence of spurious inertial modes (see, e.g., Le Roux (2012)) and decoupling between the nearest horizontal velocities. Although both can be controlled by lateral viscosity, the control leads to higher viscous dissipation over the triangular portions of the mesh. The hybrid meshes can be designed so that triangular cells are included only to optimally match the resolution or even absent altogether. For example, FESOM-C can be run on curvilinear meshes combining smooth changes in the shape of quadrilateral cells with smoothly approximated coastlines. One can also think of meshes where

triangular patches are only used to provide transitions between quadrilateral parts of different resolution, implementing an effective nesting approach.

Many unstructured-mesh coastal ocean models were proposed recently (e.g., Casulli and Walters, 2000; Chen et al., 2003; Fringer et al., 2006; Zhang and Baptista, 2008; Zhang et al., 2016). It will take some time for FESOM-C to catch them up as concerns functionality. The decision on the development of FESOM-C was largely motiated by the desire to fit in the existing

modeling infrastructure (mesh design, analysis tools, input-output organization), and not by any deficiency ofexisting models. The real work load was substantially reduced through the use or modification of the existing FESOM2 routines.

We formulate the main equations and their discretization in the three following sections. Section 5 presents results of test simulations, followed by discussion and conclusions.

## 2  Model formulation

### 2.1  The Governing Equations

We solve standard primitive equations in the Boussinesq, hydrostatic and traditional approximations. The solution is sought in the domain $\widehat{Q} = Q \times [0, t_f]$, where $t_f$ is the time interval. The boundary $\partial Q$ of domain $Q$ is formed by the free water surface, the bottom, and lateral boundaries, composed of the solid part $\partial Q_1$ and the open boundary $\partial Q_2$, $Q = \{x, y, z; x, y \in$





$\Omega, -h(x,y) \leq z < \zeta(x,y,t)\}, 0 \leq t \leq t_f$. Here $\zeta$ is the surface elevation and $h$ the bottom topography. We seek the vector of unknown $q = (\mathbf{u}, w, \zeta, T, S)$, where $\mathbf{u} = (u, v)$ is the horizontal velocity, $w$ the vertical velocity, $T$ the potential temperature and $S$ the salinity,

$$\frac{\partial \mathbf{u}}{\partial t} + \frac{\partial}{\partial x_i}(\mathbf{u}u_i) + \frac{\partial}{\partial z}(\mathbf{u}w) + \frac{1}{\rho_0}\nabla p + f\mathbf{k} \times \mathbf{u} = \frac{\partial}{\partial z}\vartheta\frac{\partial \mathbf{u}}{\partial z} + \nabla \cdot (K\nabla)\mathbf{u}, \tag{1}$$

$$\nabla \cdot \mathbf{u} + \frac{\partial}{\partial z}w = 0, \tag{2}$$

$$\frac{\partial p}{\partial z} = -g\rho, \tag{3}$$

$$\frac{\partial \Theta_j}{\partial t} + \mathbf{u}_i\frac{\partial \Theta_j}{\partial x_i} + w\frac{\partial \Theta_j}{\partial z} = \frac{\partial}{\partial z}\vartheta_\Theta\frac{\partial \Theta_j}{\partial z} + \nabla(K_\Theta\nabla)\Theta_j, \tag{4}$$

here $i = 1, 2$, $x_1 = x$, $x_2 = y$, $u_1 = u$, $u_2 = v$, and summation is implied over the repeating indices $i$; $p$ is the pressure; $j = 1, 2$ with $\Theta_1 = T$, $\Theta_2 = S$ the potential temperature and salinity respectively. The seawater density is determined by the equation of state $\rho = \rho(T, S, p)$, $\rho_0$ is the reference density; $f$ is the Coriolis parameter; $k$ is the vertical unit vector; $\vartheta$ and $K$ are the coefficients of vertical and horizontal turbulent momentum exchange, respectively; $\vartheta_\Theta$ and $K_\Theta$ are the respective diffusion

15 coefficients and $g$ is the acceleration due to gravity.

Writing

$$\rho(x, y, z, t) = \rho_0 + \rho'(x, y, z, t), \tag{5}$$

where $\rho'$ density fluctuation, we obtain, integrating Eq.(3),

$$p - p_{atm} = \int_z^\zeta \rho g dz = g\rho_0(\zeta - z) + g\int_z^\zeta \rho' dz,$$

20 where $p_{atm}$ is the atmospheric pressure. The horizontal pressure gradient is expressed then as the sum of barotropic, baroclinic and atmospheric pressure gradients:

$$\rho_0^{-1}\nabla p = g\nabla\zeta + g\rho_0^{-1}\nabla I + \rho_0^{-1}\nabla p_{atm}, \ \ I = \int_z^\zeta \rho' dz. \tag{6}$$

Note that horizontal derivatives here are taken at fixed $z$.





## 2.2 Turbulent closures

The default scheme to compute the vertical viscosity and diffusivity in the system of equations (1-4) is based on the Prandtl-Kolmogorov hypothesis of incomplete similarity. According to it, the turbulent kinetic energy $b$, the coefficient of turbulent mixing $\vartheta$ and dissipation of turbulent energy $\varepsilon$ are connected as $\vartheta = l\sqrt{b}$, where $l$ is the scale of turbulence, $\vartheta_\Theta = c_\rho \vartheta$, $\varepsilon =$

$c_\varepsilon b^2/\vartheta$; $c_\varepsilon = 0.046$ (Cebeci and Smith, 1974). Prandtl's number $c_\rho$ is commonly chosen as 0.1 and sets the dependence between the coefficients of turbulent diffusion and viscosity. The equation describing the balance of turbulent kinetic energy is obtained by parameterizing the energy production and dissipation in the equation for turbulent kinetic energy $b$ as

$$\frac{\partial b}{\partial t} - \vartheta(|\mathbf{u}_z|^2 + c_\rho g \rho_0^{-1}\frac{\partial \rho}{\partial z}) + c_\varepsilon b^2/\vartheta = \alpha_b \frac{\partial}{\partial z}\vartheta\frac{\partial b}{\partial z}, \tag{7}$$

with the boundary conditions

$b|_h = B_1|\vartheta|^2, \ \vartheta b_z|_\zeta = H\gamma_\zeta u_{*\zeta}^3,$

where $H = h + \zeta$ is the full water depth, $\alpha_b = 0.73$, $B_1 = 16.6$, $\gamma_\zeta = 0.4 \cdot 10^{-3}$; $u_{*\zeta} = (\rho/\rho_a)^{1/2}u_*$ is the dynamical velocity in water near the surface, $\rho_a$ the air density, $u_*$ the dynamic velocity of water on the interface between air and water.

Equation (7) is solved iteratively in the vertical direction for the nonlinear dissipative term. It is written as

$c_\varepsilon b^2/\vartheta = (2b^{\nu+1}b^\nu - (b^2)^\nu)/\vartheta^\nu,$

where $\nu$ is the index of iterations, which are repeated untill convergence.

To determine the turbulence scale $l$ in the presence of surface and bottom boundary layers we use the Montgomery formula (Reid , 1957)

$l = \frac{\kappa}{H}Z_h Z_\zeta,$

where $Z_h = z + h + z_h$, $Z_\zeta = -z + \zeta + z_\zeta$, $\kappa \simeq 0.4$ is the von Kármán constant, $z$ the layer depth and $z_h$, $z_\zeta$ are the roughness

parameters for the bottom and free surface respectively. To remove turbulent mixing in layers that are distant from interfaces we modify the Montgomery formula by introducing the cut-off function $Z_0 = 1 - \beta_1 H^{-2}Z_h Z_\zeta$, $0 \leq \beta_1 \leq 4$ (Voltzinger, 1985)

$l = \frac{\kappa}{H}Z_h Z_\zeta Z_0.$

In addition to the default scheme, one may select a scheme provided by the General Ocean Turbulence Model (GOTM) (Burchard et al., 1999) implemented into the FESOM-C code for computing vertical eddy viscosity and diffusion for mo-

mentum and tracer equations. GOTM includes large number of well-tested turbulence models with at least one member of every relevant model family (empirical models, energy models, two-equation models, Algebraic Stress Models, K-profile parameterisations, etc) and treats every single water column independently. Essential part of GOTM is occupied by one-point second-order schemes (Umlauf and Burchard, 2005; Umlauf et al., 2005, 2007).





## 2.3 Bottom friction parametrization

The model uses either a constant bottom friction coefficient $C_d$, or it is computed through the specified bottom roughness height $z_h$. The first option is preferable if the vertical resolution everywhere in the domain does not resolve the logarithmic layer or when the vertically averaged equations are solved. In the second option the bottom friction coefficient is computed according to Blumberg and Mellor (1987) and has the following form

$$C_d = (\ln(H/z_h)/\kappa)^2.$$

It is also possible to prescribe $C_d$ or $z_h$ as a function of horizontal coordinate at the initialization step.

## 2.4 Boundary conditions

The boundary conditions for the dynamical equations (1-2) are those of no-slip on the solid boundary $\partial Q_1$,

$$\mathbf{u}|_{\partial Q_1} = 0.$$

As is well known, formultion of open boundary conditions faces difficulties. They are related to either the lack or incompleteness of information demanded by the theory, for example, on velocity components at the open boundary. Furthermore, whatever the external information, it may contradict to the solution inside the computational domain, leading to instabilities which are frequently expressed as small-scale vortex structures forming near the open boundary. The procedure reconciling the external information with the solution inside the domain becomes of paramount importance. We use two approaches. The first one is to use a function whereby advection and horizontal diffusion are smoothly tapered to zero in the close vicinity of open boundary $\partial Q_2$. Such tapering makes the equations hyperbolic at the open boundary, so that the formulation of one condition (for example, for the elevation, $\zeta|_{\partial Q_2} = \zeta_\Gamma$) is possible (Androsov et al., 1995).

The other approach is to adapt the external information. It is applied to scalar fields and will be explained further.

Note that despite simplifications, barotropic and baroclinic perturbations still may disagree at the open boundary, leading to instabilities in its vicinity. In this case an additional buffer zone is introduced with locally increased horizontal diffusion and bottom friction.

Dynamic boundary conditions on the top and bottom specify the momentum fluxes entering the ocean. Neglecting the contributions from horizontal viscosities, we write

$$\vartheta \frac{\partial \mathbf{u}}{\partial z}|_\zeta = \tau_\zeta/\rho_0, \ \ \vartheta \frac{\partial \mathbf{u}}{\partial z}|_{-h} = \tau_h/\rho_0 = C_d|\mathbf{u}_h|\mathbf{u}_h.$$

The first of them sets the surface momentum flux to the wind stress at the surface ($\tau_\zeta$), and the second one, sets the bottom momentum flux to the frictional flux at the bottom ($\tau_h$), with $\mathbf{u}_h$ the bottom velocity.

Now we turn to the boundary conditions for the scalar quantities obeying equation (4). This is a three-dimensional parabolic equation and the boundary conditions are determined by its leading (diffusive) terms. We impose the no-flux condition on the solid boundary $\partial Q_1$

$$\frac{\partial \Theta_j}{\partial n}|_{\partial Q_1} = 0.$$





The conditions at the open boundary $\partial Q_2$ are given for outflow and inflow as Barnier et al. (1995); Marchesiello et al. (2001)

$$\Theta_t + a\Theta_x + b\Theta_y = -\frac{1}{\tau}(\Theta - \Theta_\Gamma),$$

where $\Theta_\Gamma$ is the given field value, usually a climatological one or relying on data from a global numerical model or observations. If the phase velocity components $a = -\Theta_t\Theta_x/G$, $b = -\Theta_t\Theta_y/G$ and $G = [(\partial\Theta/\partial x)^2 + (\partial\Theta/\partial y)^2]^{-1}$, Raymond and Kuo (1984) show that $\Theta$ propagates out of the domain, then $\tau = \tau_0$. If it propagates into the domain, then $a$ and $b$ are set to zero and $\tau = \tau_\Gamma$, with $\tau_\Gamma \ll \tau_0$. The parameter $\tau$ is determined experimentally and commonly is from hours to days. In the FESOM-C such an adaptive boundary condition is routinely applied for temperature and salinity, yet it can also be used for any components of solution.

The vertical boundary condition for the tracer equation (4) is that of zero flux at $z = -h$ (the bottom is insulated)

$$\vartheta_\Theta \frac{\partial\Theta_j}{\partial n}\Big|_{-h} = 0, \tag{8}$$

where the derivative is in the direction normal to the surface ($i = 1, 2$, $\Theta_1 = T$, $\Theta_2 = S$) and we neglect the contributions from horizontal diffusion. At the surface the fluxes are due to the interaction with the atmosphere,

$$\vartheta_\Theta \frac{\partial T}{\partial z}\Big|_\zeta = \hat{Q}(x, y, t)/\rho_0 c_p, \tag{9}$$

$$\vartheta_\Theta \frac{\partial S}{\partial z}\Big|_\zeta = W_s S_w/\rho_0, \tag{10}$$

where $\hat{Q}$ is the heat flux, $c_\rho$ the specific heat of sea water, $W_s = E - P$ is the evaporation minus precipitation rates, and $S_w$ is the salinity of added water. In most cases $S_w = 0$. In the presence of rivers, their discharge is added either as a prescribed inflow at the open boundary in the river mouth, or as volume sources of mass, heat and momentum distributed in the vicinity of open boundary. In the first case it might create an initial shock in elevation, so the second method is safer.

## 3 Temporal discretization

As is common in coastal models, we split the fast and slow motions into, respectively, barotropic and baroclinic subsystems (Lazure and Dumas, 2008; Higdon, 2008; Gadd, 1978; Blumberg and Mellor, 1987; Deleersnijder and Roland , 1993). The reason for this splitting is that surface gravity waves (external mode) are fast and impose severe limitations on the time step, whereas the internal dynamics can be computed with a much larger time step. The time step for the external mode $\tau_{2D}$ is limited by the speed of surface gravity waves, and that for the internal mode, $\tau_{3D}$, by the speed of internal waves or advection. The ratio $M_t = \tau_{3D}/\tau_{2D}$ depends on applications, but is commonly between 10 and 30. In practice, additional limitation are due to vertical advection, or wetting and drying processes. We will further use the indices $k$ and $n$ to enumerate the internal and external time steps respectively.

The numerical algorithm passes through several stages. On the first stage, based on the current temperature and salinity fields (time step $k$) the pressure is computed from hydrostatic equilibrium equation (6) and then used to compute the baroclinic



pressure gradient $\rho_0^{-1}\nabla p = g\nabla\zeta^k + g\rho_0^{-1}\nabla I^k + \rho_0^{-1}\nabla p_{atm}$. We use an asynchronous time stepping, assuming that integration of temperature and salinity is half-step shifted with respect to momentum. The index $k$ on $I$ implies that it is centered between $k$ and $k+1$ of momentum integration. The elevation in the expression above is taken at time step $k$, which makes the entire estimate for $\nabla p$ only first-order accurate with respect to time.

5    At the second stage, the predictor values of the three-dimensional horizontal velocity are determined as

$$\tilde{\mathbf{u}}^{k+1} - \mathbf{u}^k = \tau_{3D}(-f\mathbf{k}\times\mathbf{u} - \nabla\cdot\mathbf{u}\mathbf{u} + \nabla\cdot(K\nabla\mathbf{u}))^{AB3} - \tau_{3D}\rho_0^{-1}\nabla p + \tau_{3D}\partial_z\vartheta\partial_z\tilde{\mathbf{u}}^{k+1} - \tau_{3D}\partial_z(w\mathbf{u})^{AB3},$$

here $K$ is the coefficient of horizontal viscosity, and $AB3$ implies the Adams–Bashforth third-order estimate. The horizontal viscosity operator can be made biharmonic or replaced with filtering as discussed in the next chapter.

To carry out mode splitting, we write the horizontal velocity as the sum of the vertically averaged one $\bar{\mathbf{u}}$ and the deviation thereof (pulsation) $\mathbf{u}'$:

$$\mathbf{u} = \bar{\mathbf{u}} + \mathbf{u}', \ \ \bar{\mathbf{u}} = \frac{1}{H}\int\limits_{-h}^{\zeta}\mathbf{u}\,dz, \ \ \int\limits_{-h}^{\zeta}\mathbf{u}' = 0.$$

By integrating the system (1)-(3) vertically between the bottom and surface, with regard for the kinematic boundary conditions $\partial_t\zeta_{\tau_{2D}} + \mathbf{u}\nabla\zeta = w$ on the surface and $-\mathbf{u}\nabla h = w$ at the bottom and time discretization, we get

$$(\zeta^{n+1} - \zeta^n) + \tau_{2D}\nabla(H\bar{\mathbf{u}})^{AB3} = 0,$$

$$\bar{\mathbf{u}}^{n+1} - \bar{\mathbf{u}}^n = \tau_{2D}(-f\mathbf{k}\times\bar{\mathbf{u}} - \nabla\cdot\bar{\mathbf{u}}\bar{\mathbf{u}} + \nabla\cdot(K\nabla\bar{\mathbf{u}}))^{AB3} - \tau_{2D}(g\nabla\zeta)^{AM4} + \tau_{2D}(\tau_\zeta/\rho_0 - \tau_h) - \tau_{2D}R'_{3D} - \tau_{2D}g\rho_0^{-1}\nabla\bar{I}^k.$$

Here a specific version of $AB3$ is used, $\bar{\mathbf{u}}^{AB3} = (3/2+\beta)\bar{u}^k - (1/2+2\beta)\bar{u}^{k-1} + \beta\bar{u}^{k-2}$, with $\beta = 0.281105$ for stability reasons (Shchepetkin and McWilliams, 2005); $AM4$ implies the Adams–Multon estimate $\zeta^{AM4} = \delta\zeta^{n+1} + (1-\delta-\gamma-\epsilon)\zeta^n + \gamma\zeta^{n-1} + \epsilon\zeta^{n-2}$, taken with $\delta = 0.614$, $\gamma = 0.088$, $\epsilon = 0.013$ (Shchepetkin and McWilliams, 2005). In the equations above $\tau_s$,

20    $\tau_b$ are the surface (wind) and bottom stresses respectively, $\nabla\bar{I}$ is the vertically integrated gradient of baroclinic pressure. The term $R'_{3D}$ contains momentum advection and horizontal dissipation of the pulsation velocity integrated vertically

$$R'_{3D} = \frac{1}{H^k}[\int\limits_{-h}^{\zeta}\nabla\cdot\mathbf{u}'\mathbf{u}' - \int\limits_{-h}^{\zeta}\nabla\cdot(K\nabla\mathbf{u}')]^k.$$

In this expression $H^k$ is the total fluid depth at time step $k$, $H^k = h + \zeta^k$. This term is computed only on the baroclinic time step and kept constant through the integration of the internal mode.

25    The bottom friction is taken as

$$\tau_h = C_d|\bar{\mathbf{u}}|^n(\bar{\mathbf{u}}^{n+1} - \bar{\mathbf{u}}^n)/H^{n+1} + C_d|\tilde{\mathbf{u}}_h|^{k+1}\tilde{\mathbf{u}}_h^{k+1}/H^{n+1}.$$

The first part of bottom friction is needed to increase stability, while the second part estimates the correct friction, with $\tilde{\mathbf{u}}_h^{k+1}$ the horizontal velocity vector in the bottom cell on the predictor time step.





The system of the vertically averaged equations is stepped explicitly (except for the bottom friction) through $M_t$ time steps of duration $\tau_{2D}$ (index $n$), to 'catch up' the $k+1$ baroclinic time step. The update of elevation is made first, followed by the update of vertically integrated momentum equations.

At the "corrector" step, the 3D velocities are corrected to the surface elevation at $k+1$

$$\mathbf{u}^{k+1} = \frac{\triangle_i^k}{\triangle_i^{k+1}}\tilde{\mathbf{u}}^{k+1} + (\bar{\mathbf{u}}^{k+1} - \bar{\mathbf{u}}^P),$$

with $\bar{\mathbf{u}}^P = \frac{1}{H^{k+1}}\sum_{-h}^{\zeta}(\tilde{\mathbf{u}}^{k+1}\triangle_i^k)$, $i$ is the vertical index. Here $\triangle_i^k$, $\triangle_i^{k+1}$ are the thickness of the $i$-th layer calculated on respective baroclinic time steps. The layer thickness is $\triangle_i^k = \triangle_i H^k/H$. This correction removes the barotropic component of the predicted velocity and combines the result with the computed barotropic velocity. We will suppress the layer index $i$ where it is unambiguous.

The final step in the dynamical part calculates the transformed vertical velocity $w^{k+1}$ from 3D continuity equation (2). It is used in the next predictor step. Note that in the predictor step the computations of vertical viscosity are implicit.

New horizontal velocities, the so-called "filtered" ones, are used for avection of tracer. They are given by the sum of the "filtered" depth-mean and the baroclinic part of the "predicted" velocities (Deleersnijder, 1993),

$$\mathbf{u}_F^{k+1} = \frac{\triangle_i^k}{\triangle_i^{k+1}}\tilde{\mathbf{u}}^{k+1} + (\bar{\mathbf{u}}^F - \bar{\mathbf{u}}^P),$$

with $\bar{\mathbf{u}}^F = \frac{1}{M_t H^{k+1}}\sum_{n=1}^{n=M_t}(\bar{\mathbf{u}}^n H^n)$. The procedure of "filtering" removes possible high-frequency component in the barotropic velocity. It also improves accuracy for it in essence works toward centering the contribution of the elevation gradient. Once the filtered velocity is computed, the vertical velocity is updated to match it.

Then, the equation for temperature will be computed in the conservation form:

$$\triangle^{k+1}T^{k+1} = \triangle^k T^k - \tau_{3D}[\nabla \cdot (\mathbf{u}_F^{k+1}\triangle^k T^*) + w_{Ft}T_t^* - w_{Fb}T_b^*] + D + \tau_{3D}R + \tau_{3D}C,$$

where $D$ combines the terms related to diffusion, $w_{Ft}, w_{Fb}, T_t, T_b$ are the vertical transport velocity and temperature on top and bottom of the layer, $T^*$ will be computed trough second-order Adams–Bashforth (AB) time stepping method. $R$ is the boundary termal flux (either from surface, or due to river discharge). The last term in the equation above is

$$C = T\frac{\partial\triangle_i}{\partial t} = -T(\nabla(\triangle_i\mathbf{u}_F^{k+1}) + \partial_z(\triangle_i w_F^{k+1})).$$

Its two constituents combine to zero because of continuity. Keeping this term makes sense if computation of advection are split
into horizontal and vertical substeps. The salinity is treated similarly.

In simulations of coastal dynamics it is often necessary to simulate flooding and drying events. Explicit time stepping methods of solving the external mode are well suited for this (Luyten et al., 1999; Blumberg and Mellor, 1987; Shchepetkin and McWilliams, 2005). The algorithm to account for wetting and drying will be presented in the next section. We only note that computations are performed on each time step of the external mode.





## 4 Spatial discretization

In the finite-volume method, the governing equations are integrated over control volumes and the divergence terms, by virtue of the Gauss theorem, are expressed as the sums of respective fluxes through the boundaries of control volumes. For the cell-vertex discretization the scalar control volumes are formed by connecting cell centroids with the centers of edges, which gives the so-called median-dual control volumes around mesh vertices. The vector control volumes are the mesh cells (triangles or quads) themselves, as schematically shown in Fig. 1.

The basic structure to describe the mesh is the array of edges given by their vertices $v_1$ and $v_2$, and the array of two pointers $c_1$ and $c_2$ to the cells on the left and on the right of the edge. There is no difference between triangles, quads or hybrid meshes in the cycles which assemble fluxes. Quads and triangles are described through four indices to vertices forming them; in the case of triangles the fourth index equals the first one. The treatment of triangles and quads differs slightly in computations of gradients as detailed below. We will use symbolic notation: $e(c)$ for the list of edges forming cell $c$, $e(v)$ for the list of edges connected to vertex $v$, $v(c)$ for the list of vertices defining cell of element $c$.

In the vertical direction we introduce a $\sigma$-coordinate (Phillips, 1957)

$$\sigma = \frac{z + h}{h + \zeta}, \quad 1 \leq \sigma \leq 0.$$

The lower and upper horizontal faces correspond to the planes $\sigma = 0$ and $\sigma = 1$ respectively. The vertical grid spacing is defined by the selected set of $\sigma_i$. The spacing of $\sigma_i$ is horizontally uniform in present implementation (but it can be varying) and can be selected as equidistant or based on a parobolic function with high vertical resolution near surface and bottom in the vertical,

$$\sigma_i = -(\frac{i-1}{N-1})^\varrho + 1,$$

where $N$ is the number of vertical layers. Here $\varrho = 1(2)$ gives the uniform (parabolic) distribution of vertical layers. One more possibility to use refined resolution near bottom or surface is implemented through the formula by Burchard and Bolding (2002):

$$\sigma_i = \frac{\tanh[(L_h + L_\zeta)\frac{(N-i)}{(N-1)} - L_h] + \tanh L_h}{\tanh L_h + \tanh L_\zeta} - 1,$$

where $L_h$ and $L_\zeta$ are the number of layers near bottom and surface respectively.

The vertical grid spacing is recalculated on each baroclinic time step for the vertices, where $\zeta$ is defined. It is interpolated from vertices to cells and to edges. The vector of horizontal velocity and tracers are located in the middle of vertical layers (index $i + 1/2$), but the vertical velocity is at full layers.

### 4.1 Divergence and gradients

The *divergence operator* on scalar control volumes is computed as:

$$\int_v \nabla(\triangle \mathbf{u}) dS = \sum_{e=e(v)} [(\triangle \mathbf{u} n \ell)_l + (\triangle \mathbf{u} n \ell)_r],$$


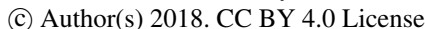


where the cycle is over edges containing vertex $v$, the indices $l$ and $r$ imply that the estimates are made on the left and right segments of the control volume boundary attached to the center of edge $e$, $\mathbf{n}$ is the outer normal and $\ell$ the length of the segment. Vectors $\mathbf{s}_l$ and $\mathbf{s}_r$ connecting the mid-point of edge $e$ with the elements on the left and on the right, we get $(\mathbf{n}\ell)_l = \mathbf{k} \times \mathbf{s}_l$ and similarly, but with the minus sign for the right element ($\mathbf{k}$ is a unit vertical vector). The mean cell values, for example layer

thickness on the cell, can be defined as $\triangle_c = \sum_{v=v(c)} \triangle_v w_{cv}$, where $w_{cv} = 1/3$ on triangles and $w_{cv} = S_{cv}/S_c$ for quads ($S_c$-cell area and $S_{cv}$-the part of it in the scalar control volume around vertex).

*Gradients of scalar* quantities are needed on cells, and are computed as:

$$\int_c \nabla \zeta dS = \sum_{e=e(c)} (\mathbf{n}\ell\zeta)_e,$$

where summation is over the edges of cell $c$, the normal and length are related to the edges, and $\zeta$ is estimated as the mean over

edge vertices.

The *gradients of velocities* on cells can be needed for computation of viscosity and momentum advection term. They are computed through the least squares fit based on the velocities on neighboring cells.

$$\pounds = \sum_{n=n(c)} (u_c - u_n - (\alpha_x, \alpha_y)\mathbf{r}_{cn})^2 = \min.$$

Here $\mathbf{r}_{cn} = (x_{cn}, y_{cn})$ is the vector connecting the center of $c$ to that of its neighbor $n$. Their solution can be reformulated in

terms of two sets of two matrix (computed once and stored) $a_{cn}^x = (x_{cn}Y^2 - y_{cn}XY))/d$ and $a_{cn}^y = (y_{cn}X^2 - x_{cn}XY))/d$, acting on velocity differences and returning the derivatives. Here $d = X^2Y^2 - (XY)^2$, $X^2 = \sum_{n=n(c)} x_{cn}^2$, $Y^2 = \sum_{n=n(c)} y_{cn}^2$ and $XY = \sum_{n=n(c)} x_{cn}y_{cn}$.

## 4.2 Momentum advection

We implemented two options for horizontal momentum advection in the flux form. The first one is the linear reconstruction

upwind, based on cell control volumes. The second one is central and is based on scalar control volumes, with subsequent averaging to cells. In the upwind implementation we write

$$\int_c \nabla(\mathbf{u}\triangle\mathbf{u})dS = \sum_{e=e(c)} (\mathbf{u}\mathbf{n}\ell\triangle\mathbf{u})_e.$$

For edge $e$, linear velocity reconstructions on the elements on its both sides are estimated at the edge center. One of the cells is $c$, and let $n$ be its neighbor across $e$. The respective velocity estimates will be denoted as $\mathbf{u}_{ce}$ and $\mathbf{u}_{ne}$ and upwind will be

written in form $2\mathbf{u} = \mathbf{u}_{ce}(1 + \text{sgn}(\mathbf{un})) + \mathbf{u}_{ne}(1 - \text{sgn}(\mathbf{un}))$, where $\mathbf{un} = \mathbf{n}((\mathbf{u}_{ce} + \mathbf{u}_{ne})/2$.

The other form is adapted from Danilov (2012). It provides additional smoothing for momentum advection by computing flux divergence for larger control volumes. In this case we first estimate the momentum flux term on scalar control volumes,

$$\int_v \nabla(\mathbf{u}\triangle\mathbf{u})dS = \sum_{e=e(v)} [(\mathbf{u}\mathbf{n}\ell\triangle\mathbf{u})_l + (\mathbf{u}\mathbf{n}\ell\triangle\mathbf{u})_r].$$

The notation here follows that for the divergence. No velocity reconstruction is involved. These estimates are then averaged

to the centers of cells. In both variants of advection form the fluid thickness is estimated at cell centers.





### 4.3 Tracer advection

Horizontal advection and diffusion terms are discretized explicitly in time. Three advection schemes have been implemented. The first two are based on linear reconstruction for control volume and are therefore second-order. The linear reconstruction upwind scheme and Miura scheme Miura (2007) differ in the implementation of time stepping. The first of them needs the

Adams-Bashforth method to be the second-order with respect to time. The scheme by Miura reaches this by estimating the tracer at a point displaced by $\mathbf{u}\tau_{3D}/2$. In both cases a linear reconstruction of tracer field for each scalar control volume is performed,

$$\Theta_R(x,y) = \Theta_0(x_v,y_v) + \Theta_x(x-x_v) + \Theta_y(y-y_v),$$

where $\Theta_0$ is the tracer value at vertex, $\Theta_x$ and $\Theta_y$ are the gradients averaged to vertex locations, and $x_v, y_v$ the coordinates of

vertex $v$. The fluxes for scalar control volume faces associated to edge $e$ are computed as

$$\sum_{e=e(v)} ([(\mathbf{u}\mathbf{n}\ell\triangle\Theta_R)_l + (\mathbf{u}\mathbf{n}\ell\triangle\Theta_R)_r]).$$

The estimate of tracer is made at the mid-points of the left and right segments, and at points displaced by $\mathbf{u}\tau_{3D}/2$ from them respectively.

The third approach used in the model is based on the gradient reconstruction. The idea of this approach is to estimate

the tracer at mid-edge locations by a linear reconstruction using the combination of centered and upwind gradients $\Theta_e^{\pm} = \Theta_{v_i} \pm \ell_e(\nabla\Theta)_e^{\pm}/2$, $i = 1, 2$ are the indices of edge vertices, and gradients are computed as

$$(\nabla\Theta)_e^+ = \frac{2}{3}(\nabla\Theta)^c + \frac{1}{3}(\nabla\Theta)^u \quad \text{and} \quad (\nabla\Theta)_e^- = \frac{2}{3}(\nabla\Theta)^c + \frac{1}{3}(\nabla\Theta)^d,$$

here, the upper index $c$ means centered estimates, and $u$, $d$ imply the estimates on the up- and down- edge cells.

The advective flux of scalar quantity $\Theta$ through the face of the scalar volume $(Q_e = [(\mathbf{u}\mathbf{n}\triangle)_l + (\mathbf{u}\mathbf{n}\ell\triangle)_r])$ associated with

edge $e$ which leaves the control volume $\nu_1$ is

$$Q_e\Theta_e = \frac{1}{2}Q_e(\Theta_e^+ + \Theta_e^-) + \frac{1}{2}(1-\gamma)|Q_e|(\Theta_e^+ + \Theta_e^-),$$

$\gamma$ is the parameter controlling the upwind dissipation. Taking $\gamma = 0$ give the third-order upwind method whereas $\gamma = 1$ gives the centered fourth-order estimate.

A quadratic upwind reconstruction is used in the vertical with the flux boundary conditions on surface (9) and (10) and zero

flux at the bottom. Other options for horizontal and vertical advection, including limiters, will be introduced in future.

The advection schemes are coded so that their order can be reduced toward the first-order upwind for very thin water layer to increase stability in the presence of wetting and drying.





### 4.4 Viscosity and filtering

Consider the operator $\nabla A \nabla \mathbf{u}$. Its computation follows the rule:

$$\int_c \nabla A \nabla \mathbf{u} dS = \sum_{e=e(c)} A\ell(\mathbf{n}\nabla\mathbf{u})_e$$

The estimate of velocity gradient on edge $e$ is symmetrized, following the standard practice, over the values on neighboring
cells. The consequence of this symmetrization is that on regular meshes (formed of equilateral triangles or rectangular quads)
the information from the nearest neighbors will be lost. Any irregularity in velocity on the nearest cells will not be penalized.
Although unfavorable for both quads and triangles, it has further reaching implications for the latter: it cannot efficiently
remove the decoupling between the nearest velocities which may occur for triangular cells. This fact is well known, and the
modification of the scheme above that improves coupling between the nearest neighbors, consists in using the identity

$$\mathbf{n} = \mathbf{r}_{cn}/|\mathbf{r}_{cn}| + (\mathbf{n} - \mathbf{r}_{cn}/|\mathbf{r}_{cn}|),$$

where $\mathbf{r}_{cn}$ is the vector connecting the centroid of cells $c$ and $n$. The derivative in the direction of $\mathbf{r}_{cn}$ is just the difference
between the neighboring velocities divided by the distance, which is explicitly used to correct $\mathbf{n}\nabla\mathbf{u}$. It is easy to show that on
rectangular quads or equilateral triangles ($n$ and $\mathbf{r}_{cn}$ are collinear) the second term of the expression above will disappear. This
is the harmonic discretization and a biharmonic version is obtained by applying the procedure twice.

A simpler algorithm is implemented to control grid-scale noise in the horizontal velocity. It consists in adding to the right
hand for the momentum equation (2D and 3D flow) a term coupling the nearest velocities,

$$F_c = -(\frac{1}{\tau_f})\sum_{n(c)}(\mathbf{u}_n - \mathbf{u}_c),$$

where $\tau_f$ a time scale selected experimentally. On regular meshes this term is equivalent to the Laplacian operator. On general
meshes it deviates from the Laplacian, yet after some trivial adjustments it warrants momentum conservation and energy
dissipation.

### 4.5 Wetting and drying algorithm

For modeling wetting and drying we use the method proposed by Stelling and Duinmeijer (2003). The idea of this method is
to accurately track the moving shoreline by employing the upwind water depth in the flux computations. The criterion for a
vertex to be wet or dry is taken as:

$$\begin{cases} \text{wet,} & \text{if} \quad D_{wd} = h + \zeta + h_l > D_{min} \\ \text{dry,} & \text{if} \quad D_{wd} = h + \zeta + h_l \leq D_{min}, \end{cases}$$

where $D_{min}$ is the critical depth and $h_l$ is the bathymetric land height. Each cell is treated as:

$$\begin{cases} \text{wet,} & \text{if} \quad D_{wd} = \min h_{v(c)} + \max \zeta_{v(c)} > D_{min} \\ \text{dry,} & \text{if} \quad D_{wd} = \min h_{v(c)} + \max \zeta_{v(c)} \leq D_{min}, \end{cases}$$





where $h_v$ and $\zeta_v$ are the depth and sea surface height at the vertices $v(c)$ of the cell $c$. When a cell is treated as dry, the velocity at its center is set to zero and no volume flux passes through the boundaries of scalar control volumes inside this cell.

## 5  Numerical simulations

In this section we present the results of two model experiments. The first of them considers tidal circulation in the Sylt–Rømø

Bight. This area has a complex morphometry with big zones of wetting/drying and large incoming tidal waves. In this case our intention is to test functioning of meshes of various kind. The second experiment simulates a South-East part of the North Sea. For this area, annual simulation of barotropic-baroclinic dynamics with realistic boundary conditions on open and surface boundaries is carried out and compared to observations. We note that a large number of simpler experiments, including those where analytical solutions are known, were carried out in the course of model development, to test and tune the model accuracy

and stability. Lessons learned from them where taken into account. We omit their discussion in favor of realistic simulations.

### 5.1  Sylt-Rømø experiment

To test the code sensitivity to the type of grid and grid quality we computed barotropic tidally driven circulation in the Sylt-Rømø Bight in the Wadden Sea.

It is a popular area for experiments and test cases (e.g. Lumborg and Pejrup, 2005; Ruiz-Villarreal et al., 2005; Burchard

et al., 2008; Purkiani et al., 2014). The Sylt and Rømø islands are connected to the mainland by artificial dams, creating a relatively small semi-enclosed bight with a circulation pattern well-known from observations and modeling (e.g. Becherer et al., 2011; Purkiani et al., 2014). It is a tidally energetic region with the water depth down to 30 m, characterized by wide intertidal flats and a rugged coastline. Water is exchanged with the open sea through a relatively narrow (up to 1.5 km wide) and deep (up to 30 m) tidal inlet Lister Dyb. The bathymetry data for the area was provided by H. Burchard and is presented in

Fig 2.

We constructed three different meshes (Fig. 2) for our experiments. The first one is a nearly regular quadrilateral mesh, complemented by triangles that straighten the coastline (MESH-1). Its spatial resolution is 200 m. The second mesh is purely triangular (MESH-2) with resolution varying from ∼820 to ∼90 m. The third mesh was generated by the Gmsh mesh generator (Geuzaine and Remacle, 2009) and includes 34820 quads and 31 triangles with the minimum cell size of 30 m and maximum

size of ∼260 m (MESH-3). All meshes have 21 non-uniform sigma layers in the vertical direction (refined near the surface and bottom). The wetting/drying option is turned on. We apply the $k - \epsilon$ turbulence closure model with transport equations for the turbulent kinetic energy and the turbulence dissipation rate using GOTM library. The second-moment closure is represented by algebraic relations suggested by Cheng et al. (2002). The experiment is forced by prescribing elevation due to $M_2$ tidal wave at the open boundary (western and northern boundaries of the domain).

Simulations on each mesh were continued until reaching the steady state in the tidal cycle of $M_2$ wave. The last tidal period was analyzed. The quasi-stationary behavior is established already on the second tidal period. The simulated $M_2$ wave is essentially nonlinear during the tidal cycle judged by the difference in amplitude of two tidal half-cycles.





Figure 3 shows the behavior of potential and kinetic energies in the entire domain, whereas the right panels show the energies computed over the areas deeper than 1 m. The results are sensitive to the meshes, which is explained further. The smallest tidal energy is simulated on the triangular mesh (MESH-2). The reason is that with the same value of the time scale $\tau_f$ in the filter used by us in these simulations the effective viscous dissipation is much higher on a triangular mesh than on quadrilateral

meshes of similar resolution. However, the solutions on quadrilateral meshes are different too, and this time the reason is the difference in the details of representing very shallow areas on meshes of various resolution (MESH-3 is finer than MESH-1). The difference between the simulations on two quadrilateral meshes is related to the potential energy and comes from the difference in the elevation simulated in the areas subject to wetting and drying. Note that the velocities and layer thickness are small in these areas, so the difference between kinetic energies between the left and right bottom panels of Fig. 3 is small.

The average currents, sea level and residual circulation simulated on MESH-1 are presented in Fig. 4. The results of this experiment show good agreement with the previously published results of Ruiz-Villarreal et al. (2005).

An example of spectrum of level oscillations on station LIST-auf-SYLT from model results presented in Fig. 5. The amplitude of the $M_2$ wave on quad meshes (MESH-1 and MESH-3) slightly exceeds 80 cm and is a bit smaller on MESH-2. Similar behavior is seen for the second harmonics ($M_4$) expressing nonlinear effects in this region. We tried to compare model

simulation with the observations (https://www.pegelonline.wsv.de/gast/start/). For comparison, the observations were taken for the first half of January, 2018. Figure 6 presents the range of fluctuations for the whole period. As is seen, the main tidal wave $M_2$ has a smaller amplitude (about 70 cm) than in simulations. However, the high-frequency part of the spectrum is very noisy because of atmospheric loading and winds. If the analysis is performed for separate tidal cycles in cases of strong wind and no-wind, the correspondence with observation is recovered in the second case.

Of particular interest is the convergence of solution on different meshes. For comparison the solutions simulated on MESH-2 and -3 were interpolated to the MESH-1. The comparison was performed for the full tidal cycle and is shown in Fig. 7 which presents the histograms of the differences.

For the solutions on MESH-1 and MESH-3 values at more than 80% of points agree within the range of $\pm1$ cm for the elevation (the maximum of tidal wave exceeds 1 m) and within the range of $\pm1$ cm/s for the velocity (the maximum horizontal

velocity is about 120 cm/s). Thus the agreement between simulations on quadrilateral MESH-1 and MESH-3 is also maintained on a local level. The agreement becomes worse when comparing solutions on triangular MESH-2 and quadrilateral MESH-1. Here the share of points with larger deviations is noticeably larger.

Spatial patterns of the differences for elevations and velocities simulated on different meshes are presented in Figs 8 and 9 respectively. Substantial differences for the elevation are located in wetting and drying zones. This is related to the sensitivity

of the wetting and drying algorithm to the cell geometry. For the horizontal velocity the difference between the solutions is defined by the resolution of bottom topography in the most energetically active zone on the quadrilateral meshes (see residual circulation in Fig. 4). The difference between the triangular grid and quadrilateral grid has a noisy character and is seen in the regions of strongest depth gradients.



## 5.2 South-East North Sea circulation

Here we present the results of realistic simulations of circulation in the southeastern part of the North Sea. The area of simulations is limited by the Dogger Bank and Horns Rev (Denmark) on the North and border between Belgium and the Netherlands on the west. It is characterized by complex bathymetry with strong tidal dynamics (Maßmann et al., 2010; Idier et al., 2017). The

related estuarine circulation (Burchard et al., 2008; Flöser et al., 2011), strong lateral salinity and nutrient gradients and rivers plumes (Voynova et al., 2017; Kerimoglu et al., 2017) are important aspects of this area. In our simulations, the mesh consists of mainly quadrilateral cells. The mesh is constructed with the Gmsh (Geuzaine and Remacle, 2009) using the Blossom-Quad method (Remacle et al., 2012). It includes 31406 quads and only 32 triangles. The mesh resolution (defined as the distance between vertices) varies between 0.5 - 1 km in the area close to the coast and Elbe estuary, coarsening to and 4 - 5 km at the

open boundary. The mesh contains $5-\sigma$ layers in the vertical. The bathymetry from the EMODnet Bathymetry Consortium (2016) has been used. Model runs were forced by 6 hourly atmospheric data from NCEP/NCAR Reanalysis (Kalnay et al., 1996) and dayly resolved observed river runoff (Radach and Pätsch , 2007; Pätsch and Lenhart, 2011). Salinity and temperature data on the open boundary were extracted from hindcast simulations based on TRIM-NP (Weisse et al., 2015). The sea surface elevation at the open boundary was prescribed in terms of amplitudes and phase for $M_2$ and $M_4$ tidal waves derived from the

previous simulations of the North Sea (Maßmann et al., 2010; Danilov and Androsov, 2015). Data for temperature and salinity from TRIM-NP model were used to initialize model runs for one year. The results of these runs were used as initial conditions for final simulation.

The validation of simulated amplitudes and phases of $M_2$ tidal wave is presented in Fig. 10. This wave is the main tidal constituent in this region. It enters the domain at the western boundary and propagates along the coast as a Kelvin wave. The

phase field is characterized by two amphidromic points. We used the observed values from Andersen (2008) for the comparison. The simulated amplitudes are generally slightly smaller than the observed ones (Fig. 10). The deviations in amplitudes can be explained by uncertainty in model bathymetry and the use of constant bottom friction coefficient. The phases of $M_2$ wave are well reproduced by the model. We characterize its accuracy by the total vector error:

$$\mu = \frac{1}{N} \sum_{n=1}^{N} ((A_* \cos \varphi_* - A \cos \varphi)^2 + (A_* \sin \varphi_* - A \sin \varphi)^2)_n^{1/2},$$

where $A_*$, $\varphi_*$, and $A$, $\varphi_*$ are the observed and computed amplitudes and phases, respectively at $N$ stations. The total vector error is 0.24 m for 53 stations in the entire simulated domain which presents a reasonably good result for this region given the domain size. From the results of comparison it is seen that observations at some stations, such as station 7 in the open sea, differ considerably from the amplitude and phase at the close stations. The comparison will improve if such outlier stations are excluded.

To validate the simulated temperature and salinity we used data from the COSYNA data base (Baschek et al., 2016). The model can represent both seasonal changes in sea surface temperature (SST) and salinity (SSS), as well as lateral gradients



(not shown) reasonably well. The modeled and observed SSS for Cuxhaven station is presented in Fig. 11 for simulations with the Miura advection scheme.

The observations are from the station located in the mouth of the Elbe river near the coast. They are characterized by tidal amplitude in excess of 1.5 m, the horizontal salinity gradient of 0.35 PSU/km and an extended wetting and drying area around
this station. Simulation is in good agreement with tidal filtered mean SSS (Fig. 11). The model represent well the summer flood event during June - July months.

The (Fig. 12) shows the calculated surface salinity field in the part of the simulated domain at the time on June 26, 2013, in comparison with the observational data from (FerryBox (FunnyGirl) (Petersen, 2014). As can be seen from the plot, there is a high consistency of the simulated results with observational data.

## 6    Discussion

### 6.1    Triangles vs. quads: numerical performance

We examine the computational efficiency by comparing the CPU time needed to simulate 5 tidal periods of $M_2$–wave on MESH-1 and MESH-2 in the Sylt-Rømø experiment, as presented on Fig. 13. The number of vertices of the quadrilateral MESH-1 is approximately $\sim 1.13$ of that of triangular MESH-2, but the numbers of elements relate as $\sim 0.57$. We have found
that the total CPU times are in approximate ratio 1.62 (triangles/quads). The simulations were performed with the same time steps.

The 3D velocity part takes approximately the same CPU time as the computation of vertically averaged velocity and elevation (external mode). Operations on elements, which include the Coriolis and bottom friction terms as well as computations of gradients of velocity and scalars, are approximately twice cheaper on quadrilateral meshes, as expected. Computations of
viscosity and momentum transport are carried out in a cycle over edges which is 1.5 times shorter for meshes made of quadrilateral elements, which warrants a similar gain of $\sim 1.5$ in performance on quadrilateral meshes. In our simulation, the net gain was $\sim 1.62$ times on MESH-1 compared to MESH-2, even despite the fact that number of vertices is 13% larger on MESH-2. Model is stable on the quadrilateral meshes with smaller horizontal viscosity, which is also an advantage.

### 6.2    Triangles vs. quads: Open boundaries

The presence of open boundaries is a distinctive feature of regional models. The implementation of robust algorithms for the open boundary is more complicated on unstructured triangular meshes than on structured quadrilateral meshes. For example, it is more difficult to cleanly assess the propagation of perturbations toward the boundary in this case. In addition, spurious inertial modes can be excited on triangular meshes in the case of cell-vertex discretization used by us, which in practice leads to additional instabilities in the vicinity of open boundary. The ability to use hybrid meshes is very helpful in this case. Indeed,
even if the mesh is predominantly triangular, the vicinity of open boundary can be constructed of quadrilateral elements.

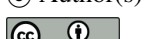



We illustrate the improvements of the dynamics in the vicinity of open boundary by simulating baroclinic tidal dynamic in an idealized channel with an underwater sill. The channel is 12 km in length and 3 km in width, with the maximum depth of 200 m near the open boundary. The sill with the height of 150 m in located in the central part of the channel. The flow is forced at the open boundaries by a tide with the period of $M_2$-wave and amplitude of 1 cm, applied in antiphase. The left part of the
channel contains denser waters than the right one.

Three meshes were used for these simulations. The first one is a quadrilateral mesh with the horizontal resolution of 200 m refined to 20 m in the vicinity of the underwater sill. The second one is a purely triangular mesh obtained from the quadrilateral mesh by splitting quads into triangles. The third mesh is predominantly triangular, but for the zones close to the open boundary where it is quadrilateral too.

Fig. 14 illustrates that at time close to the maximum of the inflow (8h 20m), a strong computational instability due to the interaction between baroclinic and barotropic flow components evolves on the right open boundary on the triangular mesh, leading eventually to the blow-up of the solution (see the left insert). However, replacing triangles in a small domain adjacent to the open boundary with quadrilateral cells we stabilize the numerical solution (see the right insert), for it allows us to cleanly handle the directions normal and tangent to the boundary.

**7  Conclusions**

We described the numerical implementation of three-dimensional unstructured-mesh model FESOM-C, relying on FESOM2 and intended for coastal simulations. The model is based on a finite-volume cell-vertex discretization and works on hybrid unstructured meshes composed of triangles and quads.

We illustrated the model performance with two test simulations.

Sylt-Rømø Bight is a closed Wadden Sea basin, characterized by a complex morphometry and high tidal activity. A sensitivity study was carried out to elucidate the dependence of simulated surface elevation and horizontal velocity on mesh type and quality. The elevation simulated in zones of wetting and drying may depend on the mesh structure, which may lead to distinctions in the simulated energy on different meshes. The total energy comparison shows that on the triangular MESH-2, having approximately the same number of vertices as MESH-1, the solution is more dissipative, for higher dissipation is generally
needed to stabilize it against spurious inertial modes.

The second experiment deals with the southeastern part of the North Sea. Computation relied on the boundary information from hindcast simulations by the TRIM-NP, and realistic atmospheric forcing from NCEP/NCAR. Modeling results agree both qualitatively and quantitatively with observations for the full period of simulation.

Future development of the FESOM-C will include coupling with the global FESOM2 (Danilov et al., 2017), addition of
monotonic high-order schemes and sea ice of FESOM2, and various modules that would increase functionality of FESOM-C.





*Code and data availability.* The version of FESOM-C used to carry out simulations reported here can be accessed from https://gitlab.dkrz.de/a270101/FESOM-NF/ after registration. The simulation results can be obtained from the authors on request.



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



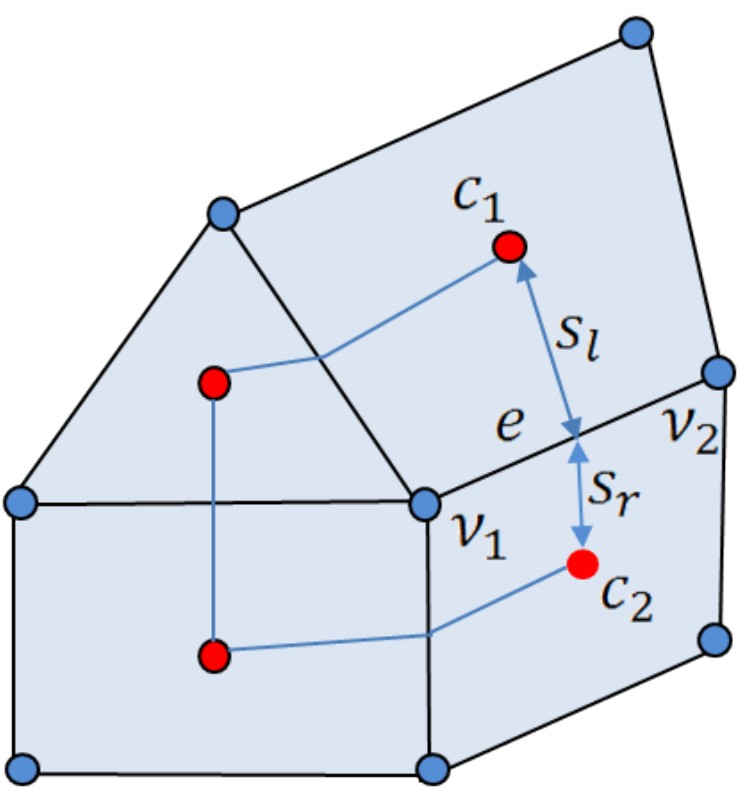

**Figure 1.** Schematic of mesh structure. Velocities are located at centroids (red circles) and elevation at vertices (blue circles). A scalar control volume associated with vertex $v_1$ is formed by connecting neighboring centroids to edge centers. The control volumes for velocity are the triangles/quads themselves. The lines passing through two neighboring centroids (e. g., $c_1$ and $c_2$) are broken in a general case at edge centers. Their fragments are described by the left and right vectors directed to centroids ($s_l$ and $s_r$ for edge $e$). Edge $e$ is defined by its two vertices $v_1$ and $v_2$ and is considered to be directed to the second vertex. It is also characterized by two elements $c_1$ and $c_2$ to the left and to the right respectively.





**Figure 2.** Top left: The bathymetry of the Sylt-Rømø Bight with the locations of two stations $P1$ and $P2$ where the comparison with GETM model is carried out; Top right: the regular quasi-quadrilateral Mesh-1 (200 m, 16089 vertices; 15578 quads and 176 triangles); Bottom left: the triangular Mesh-2 (14193 vertices and 27548 triangles); Bottom right: the irregular quadrilateral Mesh-3 (35639 vertices; 34820 quads and 31 triangles).



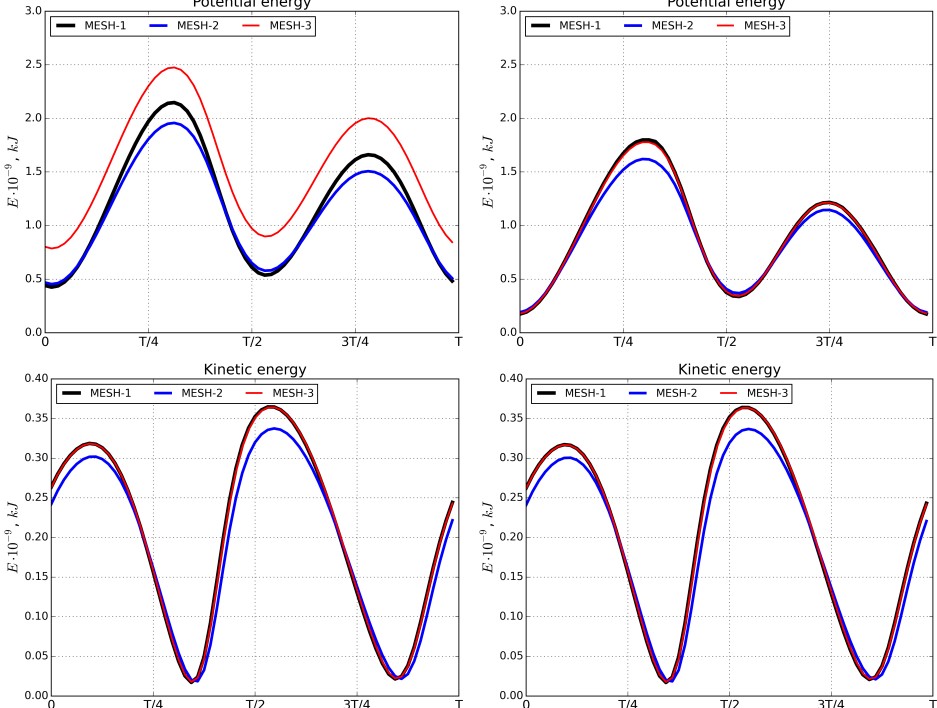

**Figure 3.** Potential and kinetic energy. Top and bottom left panels are for the total area; top and bottom right are for the area where the full depth exceeds 1 m.

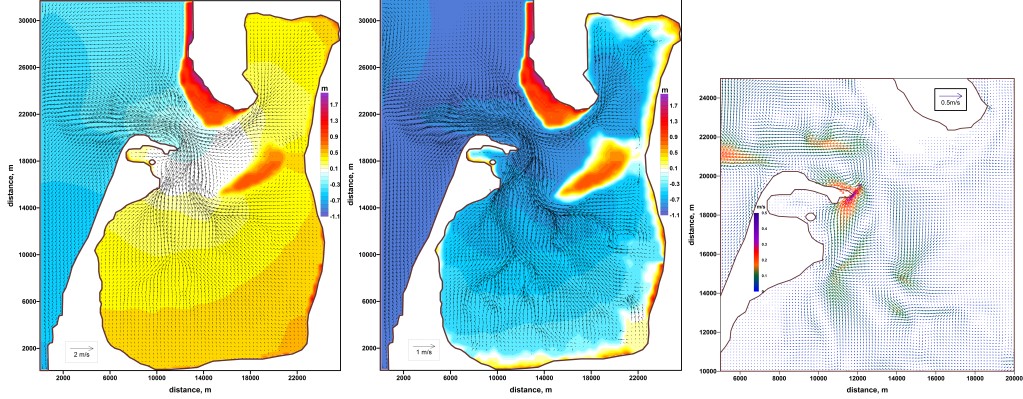

**Figure 4.** Left panel: full ebb; Middle panel: low-water; Right panel: the residual circulation. Simulation was performed on MESH-1





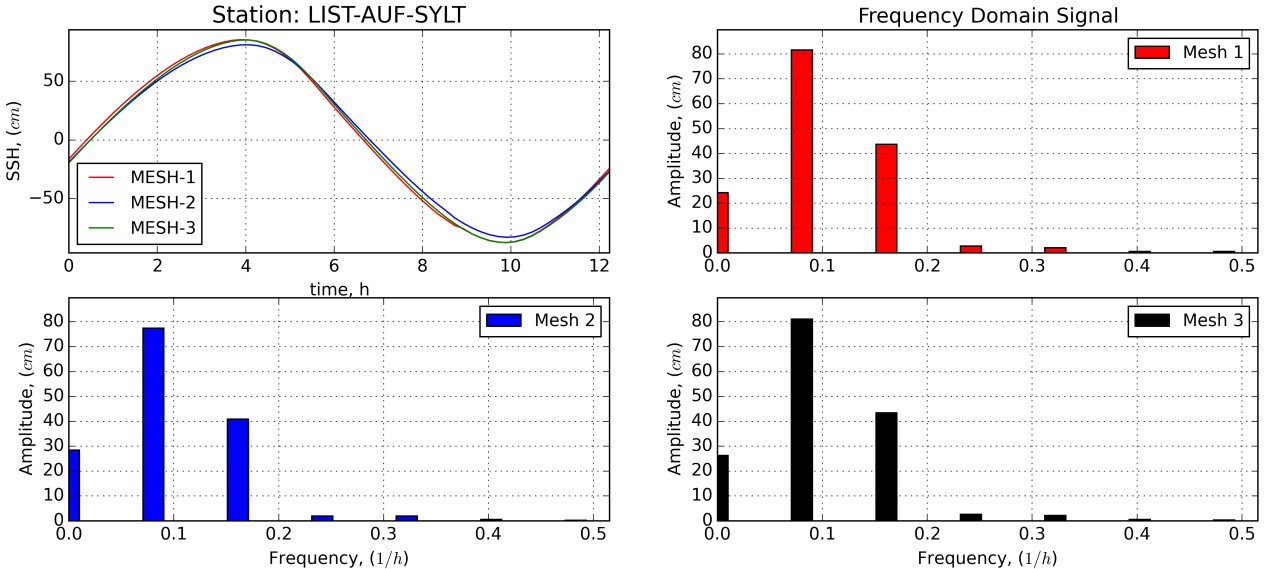

**Figure 5.** Left upper panel: SSH for one tidal period in the station List-auf-Sylt (see Fig. 2; right upper panel: spectrum of the computed $M_2$ tidal sea level at station List-auf-Sylt on MESH-1; left bottom panel: spectrum on MESH-2; right bottom panel: spectrum on MESH-3.





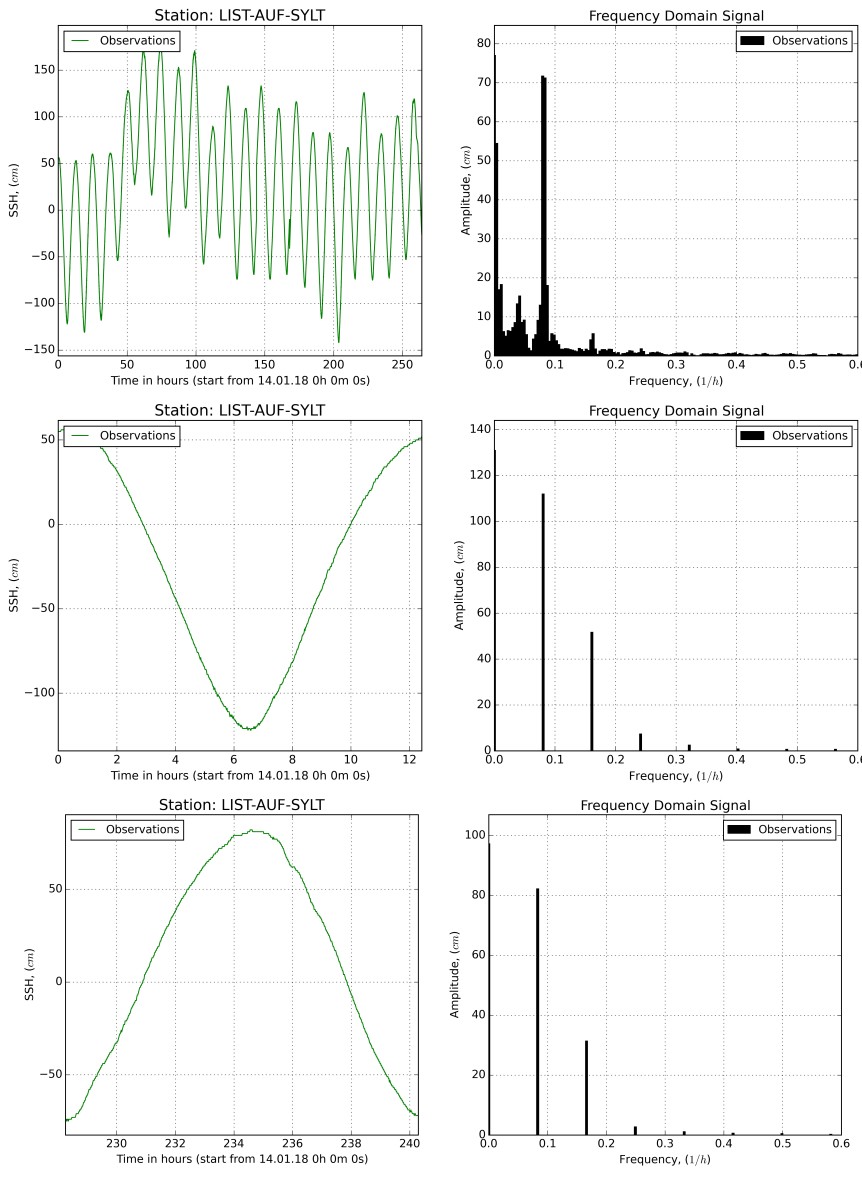

**Figure 6.** Upper panel: spectrum of the observation tidal sea level at station List-auf-Sylt (see Fig. 2 from 1 to 15 January 2018; Middle panel: spectrum of the observation SSH for one tidal period (strong wind); Bottom panel: spectrum of the observation SSH for one tidal period (no-wind).





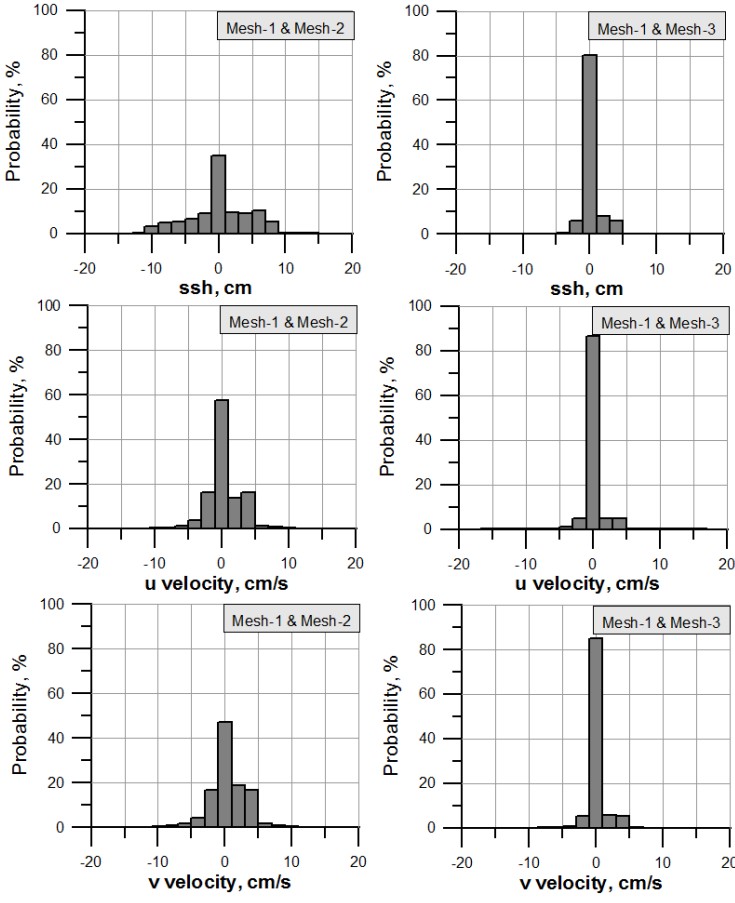

**Figure 7.** Histograms of the difference between solutions for the tidal cycle of $M_2$ wave on MESH-1 and MESH-2 (left column) and on MESH-1 and MESH-3 (right column). Top, middle and bottom rows correspond to the difference in elevation, u- and v-components of velocity respectively.

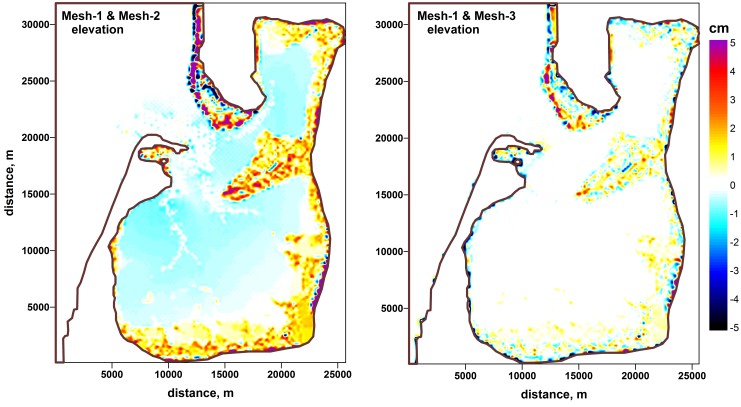

**Figure 8.** Spatial difference of the elevation for full tidal period for MESH-1 and MESH-2 (left) and for MESH-1 and MESH-3 (right).





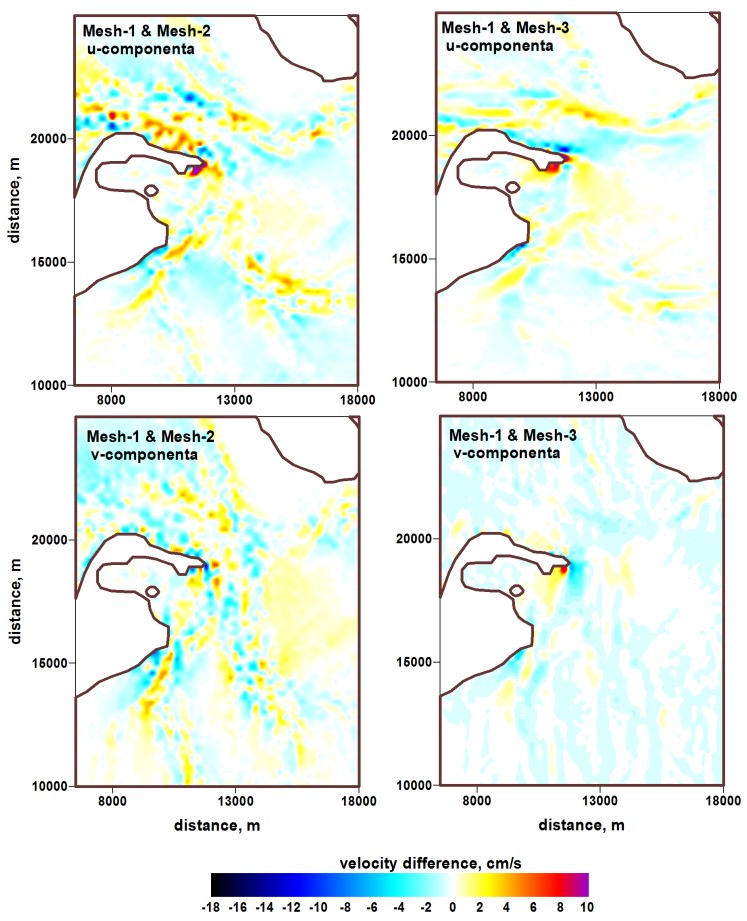

**Figure 9.** Spatial distribution of the difference between the horizontal velocities for the full tidal period of M2 wave: u- component (top row); v-component (bottom row). The differences are between MESH-1 and MESH-2 (left column) and MESH-1 and MESH-3 (right column).

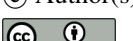



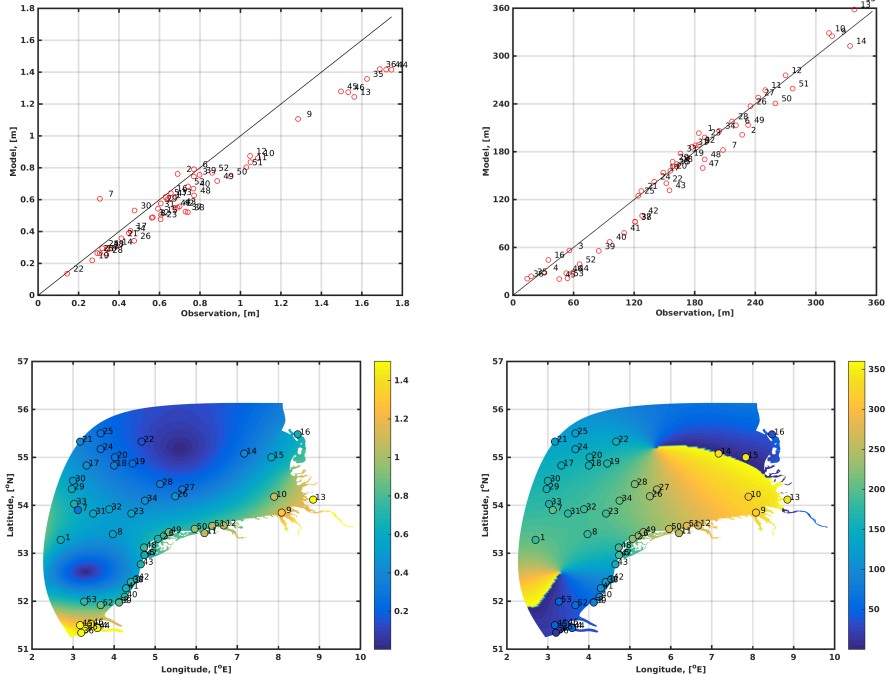

**Figure 10.** The simulated $M_2$ tidal map in the South North Sea experiment compared to observations. The amplitude is in meters (upper and bottom left) and phase (upper and bottom right) in degrees. Upper row - are model to observation graphs, the numbers correspond to stations numbers shown in bottom row. The color shows the amplitude in upper left and phase in upper right, the filled circles show the observational data. The total vector error is 0.24 m for 53 stations.





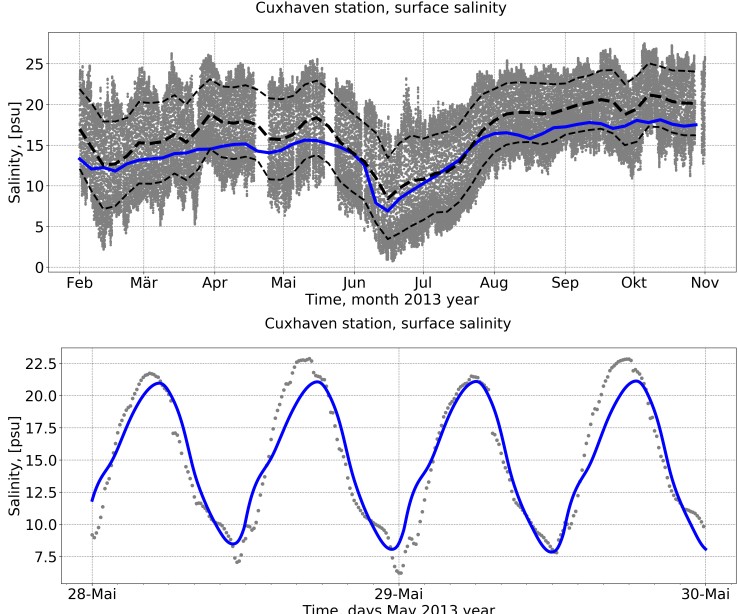

**Figure 11.** Modeled (blue line) and observed (gray dots and black lines) sea surface salinity (SSS) at the Cuxhaven station. The station is positioned at the mouth of the Elbe River between stations 9 and 13 in Fig. 10. The top panel shows 9 months of simulations, the bottom panel shows results from 2 selected day in May. The blue (modeled with the Miura advection scheme) and dashed black (observation) lines show running mean SSS with time window of 10 periods of $M_2$ tidal wave. Thin dashed black lines are one standard deviation bounds of running mean observed SSS.



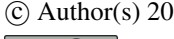

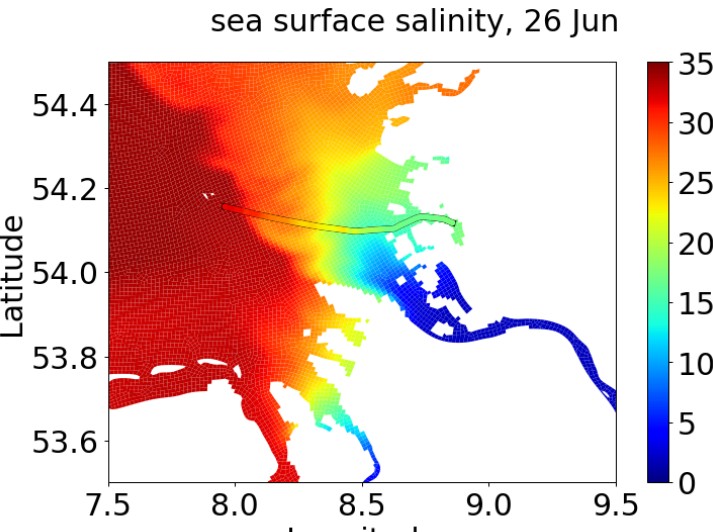

**Figure 12.** Sea surface salinity on 26 June 2013. Filled contours are model results, colored lines are observational data from FerryBox (FunnyGirl) Petersen (2014).

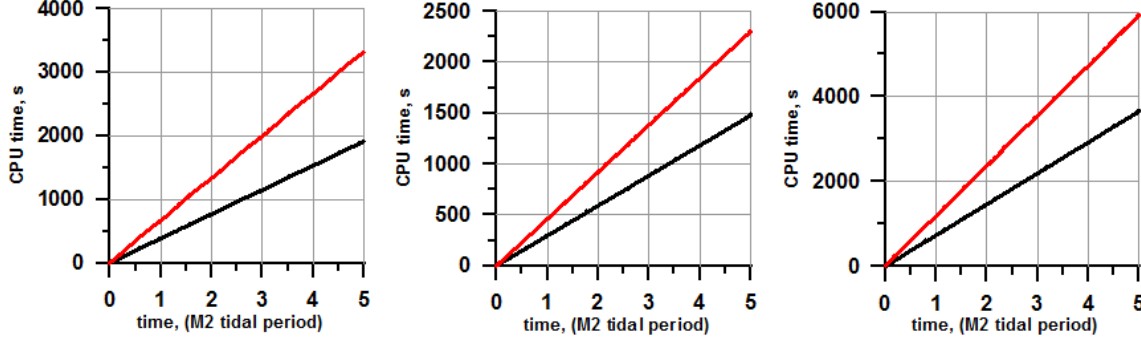

**Figure 13.** CPU time on two meshes MESH-1 and MESH-2 for Sylt-Rømø experiment. The CPU time for 3D velocity (left pannel), external mode (middle panel) and the total CPU time (right panel).





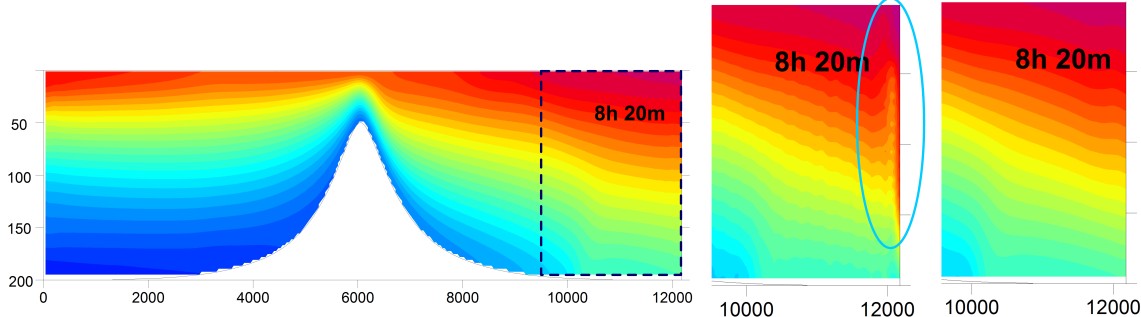

**Figure 14.** Temperature section along the channel as simulated on the quadrilaateral mesh (left panel). The two inserts show the area adjacent to the open boundary on the purely triangular mesh (left) and mesh where the vicinity of the open boundary is rendered with quads (right). The dashed rectangle shows the areaof the inserts. A numerical instability evolves on a purely triangular mesh (blue ellipse).