# Peer review of "FESOM-C v.2: coastal dynamics on hybrid unstructured meshes"

_Geoscientific Model Development, 2018_

## Short Comment (SC1) · 31 Jul 2018

Dear authors,

In my role as Executive editor of GMD, I would like to bring to your attention our Editorial version 1.1:

http://www.geosci-model-dev.net/8/3487/2015/gmd-8-3487-2015.html

This highlights some requirements of papers published in GMD, which is also available on the GMD website in the 'Manuscript Types' section:

http://www.geoscientific-model-development.net/submission/manuscript_types.html

In particular, please note that for your paper, the following requirements have not been

met in the Discussions paper:

- "The main paper must give the model name and version number (or other unique identifier) in the title."

- "All papers must include a section, at the end of the paper, entitled 'Code availability'. Here, either instructions for obtaining the code, or the reasons why the code is not available should be clearly stated. It is preferred for the code to be uploaded as a supplement or to be made available at a data repository with an associated DOI (digital object identifier) for the exact model version described in the paper. Alternatively, for established models, there may be an existing means of accessing the code through a particular system. In this case, there must exist a means of permanently accessing the precise model version described in the paper. In some cases, authors may prefer to put models on their own website, or to act as a point of contact for obtaining the code. Given the impermanence of websites and email addresses, this is not encouraged, and authors should consider improving the availability with a more permanent arrangement. After the paper is accepted the model archive should be updated to include a link to the GMD paper."

Note, that the exact Code version described in this article should be permanently accessible. Thus please consider to make the exact version, your article refers to, available via a permanent archive providing a DOI (e.g. Zenodo). Additional more information is required on how to register for the access to the FESOM code at the DKRZ gitlab.
Additionally, please add a version number identifying this version to the title of your article upon submission of the revised manuscript.

Yours,

Astrid Kerkweg

---

## Referee Comment (RC2) · Anonymous Referee #1 · 4 Oct 2018

Report: FESOM-C: coastal dynamics on hybrid unstructured meshes

General comments

This paper is a description of the coastal component of global ocean model FESOM2. I appreciate authors' honest account of aspects of model details, and think it'd be a great contribution to GMDD.

Specific comments

I give some minor comments and suggestions below

1. Eq. (7): the advection part seems missing; why?

2. Pg 5, the formula for $C_d$ should have an exponent of '-2'. More importantly, why is H

<a>Printer-friendly version</a>

<a>Discussion paper</a>

used instead of the bottom cell size? This is inconsistent with the bottom B.C. on line 25;

3. Eq. (10): if $S_w=0$ in most cases, this becomes no flux condition, and is independent of E/P values, so the model will not see their effects;

4. Section 5: all tests use only a few tidal constituents. Why not use the full tides so the model results can be compared easily with observation? Without this, I don't see how tracers can be compared for the Elbe station;

5. Fig. 11: indicate tides are not filtered out in (b)? Fig. 13: show legend for each color;

6. Pg. 17, ln 5: what's 'antiphase'?

Technical corrections

Editorial corrections: 'ofexisting' (pg 2, ln 25); Eq. (4) did not use the flux form (but the latter is used later in numerical method); Eq. (8): i should be 'j'; divergence operator should have a '.' (pg 9 ln 30), similarly on ln 25, pg. 10; far-reaching (pg 12, ln 5); (pg 12, ln 25): what's 'bathymetric land height'? pg 13, ln 10: where-> were. Fig. 5: does '1/h' mean hour^-1? Pg. 15, ln 10: daily; pg. 16, ln 20: '13% larger on '-> 'than'?

---

## Referee Comment (RC3) · Anonymous Referee #2 · 9 Nov 2018

General comments:

Modelling coastal dynamics on unstructured meshes poses - although a number of models and discretization already exists – still a number of challenges concerning stability, efficiency and performance compared to observations. The paper addresses these questions and describes a possible solution by using the proposed hybrid finite-volume cell vertex discretization. I appreciate the detailed level of description and recommend publishing it with some revisions. The overall presentation is well structured and clear, although some more explanations in the text would help the reader to follow the arguments and descriptions. The article tries to cover the mathematical description, testing and validation/real cases, which is a lot for just one article. Each part could easily be extended to be more valuable. Especially the conclusion that the re-

sults qualitatively and quantitatively agree with observation is not thoroughly supported by the presentation in the article (most of the German water level stations are missing like e.g. Cuxhaven).

Specific comments:

1.) Page 2 , Line 31: Describing the approximations as "traditional" is bit too vague. I recommend to make a reference or name it properly.

2.) Page 4, Line 10: Is there any reference or reasoning, why choosing the boundary conditions that way?

3.) Page 4, Line 14-15: This is quite short. To be complete I would expect a more detailed description about how to solve the kinetic energy equation in general or leave it completely for an appendix.

4.) Page 6, Line 6: What is taken as tau_0 and tau_gamma in the experiments?

5.) Page 8, Line 7: Why division by H in definition of delta_iˆk? Or what is Delta_i exactly?

6.) Page 8, Line 16: I don't understand the sentence "It also improves . . .. elevation gradient"

7.) Page 8, Line 19: If flux form is used in the temperature equation, it should be introduced before. It is not clear to me, why in the eq. in l. 23 the Delta only has the index i and not k.

8.) Page 9, Line27: What is meant by "full layers"? At cell surface or bottom?

9.) Page 10, Line 14-17: Maybe the reformulation and the matrices could be given in more details in an appendix?

10.) Page 10, Line 20: Define scalar control volumes

11.) Page 11: As there are several possibilities: How do you compute layer thickness

for the tracer advection?

12.) Page 12, Line 4: What is meant by "symmetrized following the standard practice" exactly? Give at least a reference.

13.) Page 12, Line 10: This is a trivial equation. Maybe there is something missing?

14.) Page 12, Line 14: Do you call it the "harmonic discretization" or is there any reference to former work, where it is properly defined or derived?

15.) Page 12, Line 18-20: The equivalence and the trivial adjustments are not obvious for me. Could you explain a bit more.

16.) Page 13, Line 8-10: It is a pity that these simpler experiments and especially the learned lessons are not published. It would advance the understanding of the problematic issues other developers may also be struggling with.

17.) Page 13ff: What values for tau_2d and tau_3d are used in the experiments (for real cases and the numerical performance test)?

18.) Page 13ff: As several discretization schemes are presented in Section 2, 3 and 4, which ones are actually used in the experiments? Otherwise present only the ones used.

19.) Page 13: What open boundary forcing is used in the Sylt-Romo experiment?

20.) Page 14: What time scale tau_f is used in the experiments? How much additional dissipation is added in comparison to other terms in momentum equations?

21.) Page 14, Line 19: Fig 5 and Fig 6: A plot of the observations at low wind conditions and of the model results would help to see the "correspondence with observation".

22.) Page 15: A figure of the South-East North Sea grid would be nice. Why only 5 sigma layers are used compared to 21 in the other experiment?

23.) Page 15: What simulation period is taken for the South-East North Sea experi-

ment? Which T&S forcing has been taken at the river Elbe input?

24.) Page 16, Line 1: What is "reasonably well"? Give statistical numbers or compare to other model results.

25.) Page 16, Line 7-9: To my opinion the Elbe fresh water plume is further north than in the observation.

26.) Page 16, Line 23: Is the viscosity smaller because less filtering has been applied on the quadrilateral mesh? Or were other parameters also changed? A table with the used parameters for each mesh and experiment would be nice.

27.) Page 17, Line 4: What is antiphase?

28.) Page 18: The code is not available for non-dkrz users (FAIR principles).

Technical corrections:

1.) When writing equations please use one line for one equation, not several equations in one line (e.g. p. 4 l. 10 or p.5 l. 25).

2.) p. 5 l. 11: formultion -> formulation

3.) p.7 l. 19: I don't' see tau_s and tau_b in the equations

4.) p.8 l. 22: termal -> thermal

5.) p.10, l. 1: Here a reference to Fig. 1 would be nice

6.) p.10, l. 3: elements = cell centers?

7.) p.10, l. 30: The information that the cell thickness is estimated at cell centers should be given before the two equations of the momentum advection

8.) p.11, l. 4: Put Miura,2007 in brackets

9.) p.11, l. 12: With left and right segments is meant s_l and s_r? Better write it and refer to Fig.1

10.) p.11, l. 20: Make reference to Fig.1 for definition of ny_1

11.) p.11, l. 25: zero flux at the bottom is Eq.8? Maybe refer to it as well?

12.) P.12, l. 13: points are collinear, vectors are parallel.

13.) p.14, l. 8: For "differences in the elevation" give reference to Fig.8.

14.) p.14, l. 16: Figure 6 -> Fig.6

15.) p.15, l. 10: write out sigma, not greek letter

16.) p.16, l. 4: Give reference for the 0.35PSU/km

17.) Fig. 2.: Check caption: no comparison with GETM was carried out, no points P1 and P2 are mentioned in the text.

18.) Fig. 4: The pictures should be bigger. It is not possible to see the current arrows and the legend. Depth is shown with respect to what? NN? Check caption: Is "full ebb" the time of maximum ebb speed? Maybe better give time after high water or low water.

19.) Fig. 6: For the middle and the bottom panel add the displayed day in the caption.

20.) Fig. 8: Check caption: "Spatial difference of the elevation" =? Spatial distribution of the elevation differences?

21.) Fig. 10: The numbers of the stations are hardly visible. Increasing the size of the pictures could help.

22.) Fig. 11: The caption needs to be rewritten because seemingly the lower panel does not show the running mean. The stations position could be shown in Fig 10.

23.) Fig. 12: Why are the dry falling areas masked out in Fig. 12? It would be nice to add a coastline in Fig. 12.

24.) Fig. 13: Add in the caption to which mesh the red and black line refer to.

---

## Author Comment (AC1) · 8 Dec 2018

Manuscript Number: **gmd-2018-112**

Article Title: **FESOM-C v.2: coastal dynamics on hybrid unstructured meshes**.

GMD

Dear Editor,

Thank you for the opportunity to answer to the issues raised by the reviewer and improve the manuscript.

In the following we will answer the comments in detail:

**Referee #1:** This paper is a description of the coastal component of global ocean model FESOM2. I appreciate authors' honest account of aspects of model details, and think it'd be a great contribution to GMDD. I give some minor comments and suggestions below.

> *Answer: We thank the reviewer for summing up the manuscript and appreciating our efforts. We try to address the remaining issues in a satisfactory manner.*

**Referee: Eq. (7): the advection part seems missing; why?**

> *Answer: This is a simplified form assuming horizontal homogeneity, so that the change of kinetic energy $db/dt$ can be written without advection terms (see for example: Monin, Jaglom, "Statistical Fluid Mechanics" v.1, 1965; Kaimal and Finnigan, "Atmospheric boundary Layer Flows", 1994). This simplification is widely used in turbulent models (see GOTM). Of course, a more general form can be used if needed.*

**Referee: Pg. 5, the formula for Cd should have an exponent of '-2'. More importantly, why is H used instead of the bottom cell size? This is inconsistent with the bottom B.C. on line 25.**

> *Answer: It our typo. It is corrected, many thanks!*

**Referee: Eq. (10): if S_w=0 in most cases, this becomes no flux condition, and is independent of E/P values, so the model will not see their effects.**

> *Answer: We added additional comments to the new version of the article. The impact of the evaporation/precipitation has been included as a volume source in the salinity and continuity equations (additional water volume provided by evaporation/precipitation).*

**Referee: Section 5: all tests use only a few tidal constituents. Why not use the full tides so the model results can be compared easily with observation? Without this, I don't see how tracers can be compared for the Elbe station.**

> *Answer: The current manuscript represents the state of our model approximately 1 year ago (we submitted it to GMD at the beginning of this year). We used mainly M2 constituent to show the model ability to represent the main features of tidal dynamics, keeping the design as simple as possible to be able to diagnose possible errors. Main*

*dynamics in this area defined by the M2 wave, as can be seen from our comparison. We now work on a manuscript analyzing longer simulations with more constituents. To show the ability of FESOM-C to work with multiple constituents we include a figure plotting the SSH at station Helgoland (Fig. 11 in our manuscript) based on results of FESOM-C for same region with 9 constituents.*

[Figure]

*As an additional analysis it would be of interest to compare dynamics of tracers in two setups with M2 only and with several constituent. However, it is beyond the scope of present manuscript were we only describe the new model and its abilities.*

**Referee: Fig. 11: indicate tides are not filtered out in (b)? Fig. 13: show legend for each color.**

*Answer: We modified the caption as follows (in the new version of the article this is Fig.12):*

*Fig. 12. Modeled (blue line) and observed (gray dots and dashed black lines) sea surface salinity (SSS) at the Cuxhaven station. The station is positioned at the mouth of the Elbe River between stations 9 and 13 in Fig. 11. The top panel shows 9 months of simulations. The bottom panel shows results from 2 selected days in May. The blue (modeled with the Miura advection scheme) and thick dashed black (observation) lines in the top panel show running mean SSS with time window of 10 periods of M2 tidal wave. Thin dashed black lines are one standard deviation bounds of running mean observed SSS on the top panel.*

**Referee: Pg. 17, ln 5: what's 'antiphase'?**

*Answer: "antiphase" means that two opposite open boundaries are shifted by 180° (by a half period) with respect to each other.*

**Technical corrections:**

Editorial corrections: 'ofexisting' (pg 2, ln 25); →*Many thanks. Corrected.*

Eq. (4) did not use the flux form (but the latter is used later in numerical method); → *Thanks. Of course, equation (4) must be written in the flux form. Corrected.*

Eq. (8): i should be 'j'; → *Thanks, it is our typo. Corrected.*

divergence operator should have a '.'
(pg 9 ln 30), similarly on ln 25, pg. 10; → *Thanks, corrected.*

Pg.12, ln. 5: far-reaching; → *Corrected.*

Pg.12, ln.25: what's 'bathymetric land height'? → *Replaced by "topography".*

Pg.13, ln.10: where-> were. → *Thanks, corrected.*

Fig.5: does '1/h' mean hourˆ-1? → *Yes. We redraw the labels on the x-axis. Now it is* $hour^{-1}$*.*

Pg.15, ln.10: daily; → *Thanks, corrected.*

Pg.16, ln.20: '13% larger on '-> 'than'? → *Thanks, corrected. It is now: …despite the fact that the number of vertices is 13% larger than on MESH-2.*

We hope our answers are satisfactory and the corrected manuscript is now adequate for publication.

With our best regards,

The authors

---

## Author Comment (AC2) · 8 Dec 2018

Manuscript Number: **gmd-2018-112**

Article Title: **FESOM-C v.2: coastal dynamics on hybrid unstructured meshes**.

GMD

Dear Editor,

Thank you for the opportunity to answer to the issues raised by the reviewer and improve the manuscript.

In the following we will answer the comments in detail:

**Referee #2:** Modelling coastal dynamics on unstructured meshes poses - although a number of models and discretization already exists – still a number of challenges concerning stability, efficiency and performance compared to observations. The paper addresses these questions and describes a possible solution by using the proposed hybrid finite-volume cell vertex discretization. I appreciate the detailed level of description and recommend publishing it with some revisions. The overall presentation is well structured and clear, although some more explanations in the text would help the reader to follow the arguments and descriptions. The article tries to cover the mathematical description, testing and validation/real cases, which is a lot for just one article. Each part could easily be extended to be more valuable. Especially the conclusion that the results qualitatively and quantitatively agree with observation is not thoroughly supported by the presentation in the article (most of the German water level stations are missing like e.g. Cuxhaven).

> *Answer: We are very grateful to the reviewer for his efforts in reading carefully our manuscript and giving us very useful and concrete comments which hopefully will help us to improve largely the paper.*

**Referee: Page 2, Line 31: Describing the approximations as "traditional" is bit too vague. I recommend to make a reference or name it properly.**

> *Answer: "Traditional" belongs to the standard text-book terminology. This model is based on the hydrostatic primitive equations in which the vertical momentum equation is reduced to a statement of hydrostatic balance and the "traditional approximation" is made in which the Coriolis force is treated approximately. The traditional approximation corresponds to taking $\Omega_y = 0$ whereby the Coriolis force becomes $f\mathbf{k}$, with $f = 2\Omega sin\varphi$, with $\mathbf{k}$ the unit vertical vector. It is valid when the aspect ratio between vertical and horizontal scales is small ($H/_L \ll 1$) (Marshall J., Hill C., Perelman L., and Adcroft A., (1997), Hydrostatic, quasi-hydrostatic, and nonhydrostatic ocean modeling, J. of Geoph. Res., v102, C3, 5733-5752). We add this reference to the article.*

**Referee: Page 4, Line 10: Is there any reference or reasoning, why choosing the boundary conditions that way?**

> *Answer: The upper boundary conditions contained an error. The total depth does not enter the second condition. It occurs only when moving to the sigma coordinate. We fixed this error.*

*The choice of these boundary conditions (BC) in the model is because they are quite well known and have a simple physical meaning. For the BC on the surface, the square of turbulent energy is proportional to the square of the dynamic velocity $u_*$. The relation between the viscosity coefficient and the dynamic velocity is established by $\gamma_\zeta$. Its value was estimated, for example, in the work of Kagan B.A et al., 1979 (Parameterization of the active layer in the model of large-scale interaction of the ocean and the atmosphere. -Meteorology and Hydrology, No. 12, pp.67-75). At the bottom, the link between the turbulent energy and the square of the velocity modulus is regulated by $B_1$ (see for example: Mellor, G. L., and T. Yamada, 1982: Development of a turbulence closure model for geophysical fluid problems. Rev. Geophys. Space Phys., 20, 851–875.)*

**Referee: Page 4, Line 14-15: This is quite short. To be complete I would expect a more detailed description about how to solve the kinetic energy equation in general or leave it completely for an appendix.**

*Answer: Some description and a reference to a detailed solution of this equation has been added to the text.*

**Referee: Page 6, Line 6: What is taken as tau_0 and tau_gamma in the experiments?**

*Answer: $\tau_0$ and $\tau_\Gamma$ were 5.0 and 0.5 [day] respectively and were chosen experimentally.*

**Referee: Page 8, Line 7: Why division by H in definition of delta_iˆk? Or what is Delta_i exactly?**

*Answer: Thanks, it was our typo error. Corrected to:*

$\Delta_i^k = \Delta_i H^k$, $\Delta_i$ is vertical grid spacing.

**Referee: Page 8, Line 16: I don't understand the sentence "It also improves … elevation gradient".**

*Answer: Filtered velocity allows us to estimate the elevation gradient in the middle of the baroclinic step (k+1 in our notation) with high accuracy. Estimation of the elevation gradient is necessary for us to correct the thickness of the layers in the transport equations.*

**Referee: Page 8, Line 19: If flux form is used in the temperature equation, it should be introduced before.**

*Answer: The equation 4 is corrected and has the flux form.*

**Referee: It is not clear to me, why in the eq. in l. 23 the Delta only has the index i and not k.**

*Answer: Thanks, it is our typo. Now in this equation:*

$(\Delta_i \mathbf{u}_F)^{k+1} + (\Delta_i w_F)^{k+1} \ldots$

**Referee: Page 9, Line27: What is meant by "full layers"? At cell surface or bottom?**

*Answer: At cell surface and bottom. In our notation, this is the index "i".*

**Referee: Page 10, Line 14-17: Maybe the reformulation and the matrices could be given in more details in an appendix?**

*Answer: In our opinion, a more detailed description of the least squares method is not required. This method is well known and quite widely used.*

**Referee: Page 10, Line 20: Define scalar control volumes.**

*Answer: The definition is present in Fig. 1. A link to this figure has been added to the text.*

**Referee: Page 11: As there are several possibilities: How do you compute layer thickness for the tracer advection?**

*Answer: The vertical grid spacing is recalculated on each baroclinic time step for the vertices, where $\zeta$ is defined. It is interpolated from vertices to cells and to edges with the weight function $w_{cv}$.*

**Referee: Page 12, Line 4: What is meant by "symmetrized following the standard practice" exactly? Give at least a reference.**

*Answer: "Symmetrized" means that the estimate on edge "e" is mean of horizontal velocity gradients computed on elements "c" and "n" (notation from article) with the common edge "e": $(\nabla u)_e = ((\nabla u)_c + (\nabla u)_c)/2$. We added this to text. (Symmetrization is needed to get non-positive kinetic energy dissipation on discrete level.)*

**Referee: Page 12, Line 10: This is a trivial equation. Maybe there is something missing?**

*Answer: The equation is indeed trivial, but not the consequences. If viscous stresses are computed using symmetric velocity gradients on edge e (see the answer above), information from the nearest neighbors will be lost in the stress divergence. Any irregularity in velocity on the nearest cells will not be penalized. The velocity gradient in the direction $r$ is the difference of velocities across the edge e divided by $|\mathbf{r}|$. Combining the estimate in direction $\mathbf{r}$ with the symmetric estimate in the direction $\mathbf{n}\text{-}\mathbf{r}$ one re-introduces coupling between the nearest velocities. This fact is well known in finite-volume literature.*

**Referee: Page 12, Line 14: Do you call it the "harmonic discretization" or is there any reference to former work, where it is properly defined or derived?**

*Answer: Harmonic (also can be called Laplacian) viscosity discretization is the common name.*

**Referee: Page 12, Line 18-20: The equivalence and the trivial adjustments are not obvious for me. Could you explain a bit more.**

*Answer:*

$$F_c = -\left(\frac{1}{\tau_f}\right)\sum_{n(c)}(\mathbf{u_n} - \mathbf{u}_c)$$

*On ideal (rectangles or equilateral triangles) meshes the expression above provides the discretization of the Laplacian operator. Indeed, by doing the Taylor series expansion around the center of c it is easy to see that $F_c \approx \left(a^2/4\right)\left(\partial_{xx} + \partial_{yy}\right)$ on triangles and 4 times that on quads, so that $\tau_f$ of about 1 day corresponds to viscosity of about $10^3\ m^2/s$ on a mesh with a side a = 10 km.*

*A reference to a more detailed description of this procedure is given.*

**Referee: Page 13, Line 8-10: It is a pity that these simpler experiments and especially the learned lessons are not published. It would advance the understanding of the problematic issues other developers may also be struggling with.**

*Answer: You are absolutely right, many test experiments could be useful in many ways. Some of the interesting test experiments have been published previously (Danilov and Androsov, 2015). Many experiments were included in our reports and presentations. In this manuscript we present one of the important experiments related to the open boundary problem.*

**Referee: Page 13ff: What values for tau_2d and tau_3d are used in the experiments (for real cases and the numerical performance test)?**

*Answer: In the Sylt-Rømø experiment we used the baroclinic (internal) time step of 7 s ($\tau_{3D}$), the barotropic ($\tau_{2D}$) one was 10 times smaller. For the South East North Sea experiment we took $\tau_{3D} = 70$ s and $\tau_{2D} = 7$ s.*

**Referee: Page 13ff: As several discretization schemes are presented in Section 2, 3 and 4, which ones are actually used in the experiments? Otherwise present only the ones used.**

*Answer: In this version of the FESOM-C model we used only one **time discretization** scheme – splitting on barotropic and baroclinic mode. For **spatial discretization** we used finite-volume method. In the model code we have different implementations for momentum advection and tracer equations. In the momentum advection we used the second order upwind scheme. For tracer equation in the South East North Sea experiment we used the Miura advection scheme. For numerical stability in both experiments we used filtering procedure.*

**Referee: Page 13: What open boundary forcing is used in the Sylt-Romo experiment?**

*Answer: The text on page 13 (Lines 28-29) describes the boundary conditions used in the model. We added the link to the source of this data:*

*The experiment is forced by prescribing elevation due to $M_2$ tidal wave at the open boundary (western and northern boundaries of the domain) provided by H. Burchard.*

**Referee: Page 14: What time scale tau_f is used in the experiments? How much additional dissipation is added in comparison to other terms in momentum equations?**

*Answer: In the both experiments we used filtering with time scale parameter equal 1 day. The contribution of this term for the quad meshes is really insignificant. On the triangular meshes, its contribution is somewhat higher (as shown by the Sylt-Rømø experiment). A more detailed answer to this question can be found in the article Danilov and Androsov, 2015.*

**Referee: Page 14, Line 19: Fig 5 and Fig 6: A plot of the observations at low wind conditions and of the model results would help to see the "correspondence with observation".**

*Answer: At this stage, the experiment for Sylt-Römö was performed without taking into account atmospheric forcing. The results shown in Fig. 6 are observation. The idea was to compare the frequency spectrum in the model simulation and observations for a period without the influence atmospheric forcing (we choose a period from observation when the sea surface height average for one tidal period was very close to zero).*

**Referee: Page 15: A figure of the South-East North Sea grid would be nice.**

*Answer: We added a new figure showing the mesh.*

[Figure]

**Referee: Why only 5 sigma layers are used compared to 21 in the other experiment?**

*Answer: We used 21 layers for baroclinic simulation and 5 layers for barotropic.*

**Referee: Page 15: What simulation period is taken for the South-East North Sea experiment? Which T&S forcing has been taken at the river Elbe input?**

*Answer: The spin-up period was one year. Final simulation was one year long too. First month of simulation was used to adjust initial conditions. Salinity in all rivers was set to 2 [psu]. The daily temperature was taken from same source as runoff. (Radach and Pätsch , 2007 ; Pätsch and Lenhart, 2011). We added to the manuscript:*

*The results of these runs were used as initial conditions for 10 months final simulation.*

**Referee: Page 16, Line 1: What is "reasonably well"? Give statistical numbers or compare to other model results.**

*Answer: Statistics of comparison to observations were added to the text:*

*To validate the simulated temperature and salinity we used data from the COSYNA data base (Baschek et al., 2016) and ICES data base (www.ices.dk). Comparison of modeled surface temperature and salinity show good Pearson correlation coefficient 0.98 and 0.9 with RMSD 1.24 and 0.98 respectively.*

**Referee:  Page 16, Line 7-9: To my opinion the Elbe fresh water plume is further north than in the observation.**

*Answer: The ferry needs approximately 2 hours to go from the land to Helgoland Island. Tidal currents in this area could reach 1 m/s. During one ferry cruise the front of fresh water plume can displace by 7 km. We used snapshots to map model salinity. For technical reasons, the 3D snapshots were output approximately every 2 hours. This introduces additional technical errors in comparison. In addition to technical issues, uncertainties in runoff and T/S parameters in rivers parametrization and coarse atmospheric forcing resolution (6h in time and 200 km in space) reduce the accuracy of simulation presented in the manuscript.  We work on the next manuscript with more detailed validation of similar experiment and reduced uncertainty in current simulations. Nevertheless, the simulation presented in the manuscript shows a sharp gradient of salinity 35 – 20 [psu] in the western part of the ferry track. The model show smaller salinity values in the eastern part of the track compared to the ferry data.*
*We redo figure in such a way that all cells of the model could be seen to answer technical question 23. We used different snapshot from the model.*

[Figure]

**Referee:  Page 16, Line 23: Is the viscosity smaller because less filtering has been applied on the quadrilateral mesh? Or were other parameters also changed? A table with the used parameters for each mesh and experiment would be nice.**

*Answer: For all meshes we used the same filter factor. The effectiveness of the filtering procedure on triangular grids was lower. This effect is examined in more detail in the paper Danilov and Androsov, 2015.*

**Referee: Page 17, Line 4: What is antiphase?**

*Answer: "antiphase" - two opposite open boundaries have a 180° phase shift (a shift of half period) each other.*

**Referee: Page 18: The code is not available for non-dkrz users (FAIR principles).**

*Answer: We have put the code described in the manuscript to the permanent data archive Zenodo with an open access, the doi is **https://doi.org/10.5281/zenodo.2085177**. The full model name and version number were added to the caption of the manuscript and to the code description at the Zenodo portal.*

**Technical corrections:**

1) When writing equations please use one line for one equation, not several equations in one line (e.g. p. 4 l. 10 or p.5 l. 25). → *Done.*

2) p. 5 l. 11: formultion -> formulation → *Thanks.*

3) p.7 l. 19: I don't' see tau_s and tau_b in the equations → *Thanks, it was our typo, changed to $\tau_\zeta$ and $\tau_h$ for surface (wind) and bottom stress respectively.*

4) p.8 l. 22: termal -> thermal → *Thanks.*

5) p.10, l. 1: Here a reference to Fig. 1 would be nice → *Done.*

6) p.10, l. 3: elements = cell centers? → *Thanks, corrected.*

7) p.10, l. 30: The information that the cell thickness is estimated at cell centers should be given before the two equations of the momentum advection → *This definition has already been given in the "Spatial discretization" section (p.9 L.25-26 old version).*

8) p.11, l. 4: Put Miura, 2007 in brackets → *Done.*

9) p.11, l. 12: With left and right segments is meant s_l and s_r? Better write it and refer to Fig.1. → *The spatial structure of the grid is given above in the spatial discretization section. The link to Figure 1 is there: "The basic structure to describe the mesh is the array of edges given by their vertices $v_1$ and $v_2$, and the array of two pointers $c_1$ and $c_2$ to the cells on the left and on the right of the edge…."*

10) p.11, l. 20: Make reference to Fig.1 for definition of ny_1 → *Done.*

11) p.11, l. 25: zero flux at the bottom is Eq.8? Maybe refer to it as well? → *This condition assumes that the temperature and salinity profile is not affected by heat flux from the bottom (the bottom is isolated). In our opinion, no additional reference is required.*

12) P.12, l. 13: points are collinear, vectors are parallel. → *Now: **n** and $\mathbf{r}_{cn}$ are collinear.*

13) p.14, l. 8: For "differences in the elevation" give reference to Fig.8. → *Done.*

14) p.14, l. 16: Figure 6 -> Fig.6 → *We have left the full word (Figure) only if the sentence begins with it. The rest was replaced by Fig.*

15) p.15, l. 10: write out sigma, not greek letter → *Done.*

16) p.16, l. 4: Give reference for the 0.35 PSU/km. → *Done:*

*(www.portal-tideelbe.de and J. Kappenberg, M. Berendt, N. Ohle, R. Riethmuller, D. Schuster and T. Strotmann. Variation of Hydrodynamics and Water Constituents in the Mouth of the Elbe Estuary, Germany. Civil Eng Res J. 2018; 4(4): 555643.)*

17) Fig. 2.: Check caption: no comparison with GETM was carried out, no points P1 and P2 are mentioned in the text. → *Thanks, corrected.*

18) Fig. 4: The pictures should be bigger. It is not possible to see the current arrows and the legend. Depth is shown with respect to what? NN? Check caption: Is "full ebb" the time of maximum ebb speed? Maybe better give time after high water or low water. → *The size of the Fig.4 is increased. The reference to bathymetry has already been indicated in the text. We also added it to Fig. 2. We saved the term "full ebb" as before (same terminology is given in the article Purkiani et al.(2014), which we used for comparison our model simulation).*

19) Fig. 6: For the middle and the bottom panel add the displayed day in the caption. → *Done.*

20) Fig. 8: Check caption: "Spatial difference of the elevation" =? Spatial distribution of the elevation differences? → *Thanks, corrected.*

21) Fig. 10: The numbers of the stations are hardly visible. Increasing the size of the pictures could help. → *We redraw figures. We also add station Cuxhaven into analyses and figure (St. 9).*

22) Fig. 11: The caption needs to be rewritten because seemingly the lower panel does not show the running mean. The stations position could be shown in Fig 10. → *We modified the caption.*

23) Fig. 12: Why are the dry falling areas masked out in Fig. 12? It would be nice to add a coastline in Fig. 12. → *We redid the figure in such a way that all cells of the model could be seen.*

24) Fig. 13: Add in the caption to which mesh the red and black line refer to. → *Thank, done.*

We hope our answers are satisfactory and the corrected manuscript is now adequate for publication.

With our best regards,

The authors

---

## Author Comment (AC3) · 8 Dec 2018

Manuscript Number: **gmd-2018-112**

Article Title: **FESOM-C v.2: coastal dynamics on hybrid unstructured meshes**.

GMD

Dear Editor,

Thank you for the opportunity to answer to the issues raised by the Executive editor of GMD.

**Executive editor of GMD:**

**"The main paper must give the model name and version number (or other unique identifier) in the title."**

**"All papers must include a section, at the end of the paper, entitled 'Code availability'. Here, either instructions for obtaining the code, or the reasons why the code is not available should be clearly stated. It is preferred for the code to be uploaded as a supplement or to be made available at a data repository with an associated DOI (digital object identifier) for the exact model version described in the paper. Alternatively, for established models, there may be an existing means of accessing the code through a particular system. In this case, there must exist a means of permanently accessing the precise model version described in the paper. In some cases, authors may prefer to put models on their own website, or to act as a point of contact for obtaining the code. Given the impermanence of websites and email addresses, this is not encouraged, and authors should consider improving the availability with a more permanent arrangement. After the paper is accepted the model archive should be updated to include a link to the GMD paper."**

> *Answer: We have put the code described in the manuscript to the permanent data archive Zenodo with an open access, the DOI is* **https://doi.org/10.5281/zenodo.2085177.** *The full model name and version number were added to the caption of the manuscript and to the code description at the Zenodo portal.*

---

## Author Comment (AC5) · 8 Dec 2018

The comment was uploaded in the form of a supplement:
https://www.geosci-model-dev-discuss.net/gmd-2018-112/gmd-2018-112-AC5-supplement.pdf

---

## Author Comment (AC6) · 8 Dec 2018

The comment was uploaded in the form of a supplement:
https://www.geosci-model-dev-discuss.net/gmd-2018-112/gmd-2018-112-AC6-supplement.pdf

---

## Author Response (AR1)

Manuscript Number: **gmd-2018-112**

Article Title: **FESOM-C v.2: coastal dynamics on hybrid unstructured meshes**.

**Main author's changes:**

We added version number (FESOM-C v.2) to the caption.

We added new co-author: Holger Brix (Institute of Coastal Research, Helmholtz-Zentrum Geesthacht, Geesthacht, Germany).

We change affiliation of the co-author Ivan Kuznetsov (now: Alfred Wegener Institute for Polar and Marine Research, Bremerhaven, Germany).

Pg.2, L.31: We add new reference to the article Marshall J., Hill C., Perelman L., and Adcroft A., (1997), Hydrostatic, quasi-hydrostatic, and nonhydrostatic ocean modeling, J. of Geoph. Res., v102, C3, 5733-5752.

Eq. (4) now in the flux form.

Pg.4, L.10: The total depth does not enter the second condition. We fixed this error.

Pg.4, L.16-19: Some description and a reference (Voltzinger, 1989) to a detailed solution of the equation 7 have been added to the text.

Pg.5, L.10: The formula for Cd now have an exponent of '-2'. Size of the bottom thickness instead of the H is $0.5h_b + z_h$.

Pg.6: We removed the Eq. 8.

Eq. (10): now $\vartheta_\Theta \frac{\partial S}{\partial z}\Big|_\zeta = 0$. We added additional comments: "The impact of the evaporation/precipitation has been included as a volume source in the salinity and continuity equations."

Pg.7, L.24: changed to $\tau_\zeta$ and $\tau_h$ for surface (wind) and bottom stress respectively.

Pg.8, L.16: Corrected description of the vertical grid spacing: " $\Delta_i^k = \Delta_i H^k$, $\Delta_i$ is vertical grid spacing."

Pg.11, L.3: We add a link to Figure 1.

Pg.12, L.15: We add some description of the "symmetrized following the standard practice": "Symmetrized" means that the estimate on edge "e" is mean of horizontal velocity gradients computed on elements "c" and "n" (notation from article) with the common edge "e": $(\nabla u)_e = ((\nabla u)_c + (\nabla u)_c)/2$. "

Pg.13, L.3: We add a reference to (Danilov and Androsov, 2015).

Pg.14, L.23: Add a link to Fig.8.

Pg. 15: We added a new figure (Fig. 10) of the South-East North Sea mesh:

[Figure]

Fig. 10. The area of South-East North Sea experiment with mesh (black lines). Red dot indicates position of the Cuxhaven station. This mesh includes 31406 quads and 32 triangles.

Pg.16, L.10-12: Statistics of comparison to observations were added to the text: "To validate the simulated temperature and salinity we used data from the COSYNA data base (Baschek et al., 2016) and ICES data base ([www.ices.dk](www.ices.dk)). Comparison of modeled surface temperature and salinity show good Pearson correlation coefficient 0.98 and 0.9 with RMSD 1.24 and 0.98 respectively."

Pg.16, L.16: Added new references: "(*[www.portal-tideelbe.de](www.portal-tideelbe.de) and J. Kappenberg, M. Berendt, N. Ohle, R. Riethmuller, D. Schuster and T. Strotmann. Variation of Hydrodynamics and Water Constituents in the Mouth of the Elbe Estuary, Germany. Civil Eng Res J. 2018; 4(4): 555643.*)

Pg.18, L.12-13. We have put the code described in the manuscript to the permanent data archive Zenodo with an open access, the DOI is **https://doi.org/10.5281/zenodo.2085177.**

Fig. 2.: Change caption and Figure.2: no points P1 and P2.

Fig. 5: We redraw the labels on the x-axis. Now it is "hour$^{-1}$".

Fig. 6: Change caption: "Upper panel: spectrum of the observation tidal sea level at station List-auf-Sylt (see Fig. 2) from 1 to 15 January 2018; Middle panel: spectrum of the observation SSH for one tidal period (strong wind: 01.01.2018); Bottom panel: spectrum of the observation SSH for one tidal period (no-wind: 14.01.2018)."

Fig. 8: Change caption: "Spatial distribution of the elevation differences…"

Fig. 11: We redraw figures (stations more visible). We also add station Cuxhaven into analyses and figure (St. 9).

Fig. 12: We modified the caption: "Figure 12. Modeled (blue line) and observed (gray dots and dashed black lines) sea surface salinity (SSS) at the Cuxhaven station. The station is positioned at the mouth of the Elbe River between stations 9 and 13 in Fig. 11. The top panel shows 9 months of simulations. The bottom panel shows results from 2 selected days in May. The blue (modeled with the Miura advection scheme) and thick dashed black (observation) lines in the top panel show running mean SSS with time window of 10 periods of M2 tidal wave. Thin dashed black lines are one standard deviation bounds of running mean observed SSS on the top panel."

Fig. 13: We redid the figure (used different snapshot from the model) in such a way that all cells of the model could be seen.

[Figure]

Figure 13. Sea surface salinity on 26 June 2013. Filled contours are model results, colored lines are observational data from FerryBox (FunnyGirl) Petersen (2014).

Fig. 14: We change caption: CPU time on two meshes MESH-1 (**black line**) and MESH-2 (**red line**) for Sylt-Rømø experiment. The CPU time for 3D velocity (left pannel), external mode (middle panel) and the total CPU time (right panel).

**Main our typo (corrected):**

Pg.2, L.25. 'ofexisting' → "of existing"

Pg.10, L.9 and Pg.11, L.2: divergence operator have a '·'

Pg.11, L.14: Reference to Miura, 2007 in brackets.

Pg.12, L.25: "**n** *and* $\mathbf{r}_{cn}$ are collinear."

Pg.13, L.9: 'bathymetric land height'? → Replaced by "topography".

Pg.17, L.4-5: "…despite the fact that the number of vertices is 13% larger **than on MESH-2**."

Pg.8 L.28: Now in this equation: $(\Delta_i \mathbf{u}_F)^{k+1} + (\Delta_i w_F)^{k+1}$…

Pg.11, L.10: elements → cell centers.

[revised manuscript text omitted]

---

## Author Response (AR2)

Manuscript Number: **gmd-2018-112**

Article Title: **FESOM-C v.2: coastal dynamics on hybrid unstructured meshes**.

GMD

Dear Editor,

We made the required corrections in the manuscript and answered some of the reviewer's questions.

**Referee #2:** I would like to thank the authors for the additional information given in the answers. The manuscript is an interesting description of the model mathematical background and gives insights and possible solutions to common numerical problems of unstructured mesh models. But still the results do not convince me with respect to the validation in the realistic cases of the Sylt-Romo and South-East North Sea circulation experiments. The results seem to depend stronger on the applied forcing (sensitivity to open boundary, river input, etc) than on the numerical mesh (see differences in Fig 7 compared to over- respectively underestimation of the tidal amplitude maybe due to the different open boundary forcing in the two experiments?). The velocity differences between the meshes do not exceed 5 cm/s (mostly less than 1 cm/s). So the question is, what is absolute velocity and the natural variability of the currents in these areas and how does this difference compare to velocity changes due to uncertainties in the forcing, parametrizations and bathymetry. Are there still improvements in the numerical schemes required? The salinity and temperature comparison with observation could be more detailed, because a RMSD of 1.24 °C is quite common for ocean models in this region. The gradients seem to be stronger than observed. So I recommend setting up the model with better meteorology and open boundary forcing (including a surge model), tune it and then perform a more detailed validation with observation. Especially the storm surge heights have not been analyzed at all. (The sea surface height average for one tidal period could be zero, although there is atmospheric influence, e.g. during permanent easterly winds lowering the sea level in the German Bight). The question is, what the models purpose is, so what are the important variables to analyse. The article has improved with the corrections and I would suggest accepting the manuscript as is (maybe do typo corrections: p.9 l.2 trough -> through and p.16 l.5 A, phi_star -> A, phi). Thanks for making code available in open source.

*Answer: We thank the reviewer for his efforts in reading carefully our manuscript and summing up the manuscript and appreciating our efforts.*

**Referee: The results seem to depend stronger on the applied forcing (sensitivity to open boundary, river input, etc) than on the numerical mesh (see differences in Fig 7 compared to over- respectively underestimation of the tidal amplitude maybe due to the different open boundary forcing in the two experiments?).**

> *Answer:* In coastal models, the result is strongly dependent on boundary information and less from the mesh. On all tested meshes, for the *Sylt-Römö* experiment, we set identical boundary information. The conclusions of our analysis suggest that the solution on triangular meshes is more dissipative (*effectiveness of the filtering procedure on triangular grids was lower, Danilov and Androsov, 2015.*) and, if possible, it is desirable to use a meshes consisting of quads.

**Referee: The velocity differences between the meshes do not exceed 5 cm/s (mostly less than 1 cm/s). So the question is, what is absolute velocity and the natural variability of the currents in these areas and how does this difference compare to velocity changes due to uncertainties in the forcing, parametrizations and bathymetry.**

Answer: Variability in velocity fields is of interest to us only in terms of the convergence of solutions on meshes of various configurations.

**Referee: Are there still improvements in the numerical schemes required?**

*Answer: Of course, the numerical scheme will have some changes in the future. The main ones will be associated with vertical approximation of the area and wetting and drying parametrization.*

**Referee: The salinity and temperature comparison with observation could be more detailed, because a RMSD of 1.24 °C is quite common for ocean models in this region. The gradients seem to be stronger than observed. So I recommend setting up the model with better meteorology and open boundary forcing (including a surge model), tune it and then perform a more detailed validation with observation. Especially the storm surge heights have not been analyzed at all. (The sea surface height average for one tidal period could be zero, although there is atmospheric influence, e.g. during permanent easterly winds lowering the sea level in the German Bight).**

*Answer: We work on the next manuscript with more detailed validation of similar experiment and reduced uncertainty in current simulations.*

**Technical corrections:**

1) p.9, l.2: *trough* → **through**. *Thanks, corrected.*

2) p.16, l.5: *A, $\varphi_*$* → ***A, $\varphi$***. *Thanks, corrected.*

With our best regards,

The authors